# RIPK1 regulates starvation resistance by modulating aspartate catabolism

Xinyu Mei[1,2,4], Yuan Guo[1,3,4], Zhangdan Xie[1,3,4], Yedan Zhong[1], Xiaofen Wu[1,3], Daichao Xu[1], Ying Li [1], Nan Liu[1] & Zheng-Jiang Zhu [1✉]

RIPK1 is a crucial regulator of cell death and survival. *Ripk1* deficiency promotes mouse survival in the prenatal period while inhibits survival in the early postnatal period without a clear mechanism. Metabolism regulation and autophagy are critical to neonatal survival from severe starvation at birth. However, the mechanism by which RIPK1 regulates starvation resistance and survival remains unclear. Here, we address this question by discovering the metabolic regulatory role of RIPK1. First, metabolomics analysis reveals that *Ripk1* deficiency specifically increases aspartate levels in both mouse neonates and mammalian cells under starvation conditions. Increased aspartate in *Ripk1⁻/⁻* cells enhances the TCA flux and ATP production. The energy imbalance causes defective autophagy induction by inhibiting the AMPK/ULK1 pathway. Transcriptional analyses demonstrate that *Ripk1⁻/⁻* deficiency downregulates gene expression in aspartate catabolism by inactivating SP1. To summarize, this study reveals that RIPK1 serves as a metabolic regulator responsible for starvation resistance.

[1] Interdisciplinary Research Center on Biology and Chemistry, Shanghai Institute of Organic Chemistry, Chinese Academy of Sciences, Shanghai 200032, P. R. China. [2] Center for Clinical Research and Translational Medicine, Yangpu Hospital, Tongji University School of Medicine, Shanghai, P.R. China. [3] University of Chinese Academy of Sciences, Beijing 100049, P. R. China. [4]These authors contributed equally: Xinyu Mei, Yuan Guo, Zhangdan Xie. ✉email: jiangzhu@sioc.ac.cn

Receptor-interacting protein kinase-1 (RIPK1) is a master regulator of cell survival and death[1–4] and mediates FADD/caspase-8-dependent apoptosis and RIPK3-dependent necrosis[5,6]. In mice, genetic ablation of cell death-related genes mostly leads to prenatal death or embryonic defects (e.g., $Fadd^{-/-}$ mice)[7]. However, $Ripk1^{-/-}$ mice appear almost normally in the prenatal period while exhibit postnatal lethality with extensive cell death within 1–3 days after birth[8–12]. Although $Ripk1$ knockout rescues the prenatal death of $Fadd^{-/-}$ mice, $Ripk1^{-/-}$ $Fadd^{-/-}$ mice still die in a short time after birth[9]. This demonstrates that RIPK1 specifically promotes survival in the neonatal period. At birth, the transplacental nutrient supply is suddenly interrupted, and neonates face severe nutritional deficiency. Neonates adapt to this severe starvation circumstance by activating autophagy to maintain the supply of amino acids and metabolic homeostasis[13]. Similar to $Ripk1^{-/-}$ mice, genetic ablation of autophagy machines, such as $Atg5$ in mice, also causes postnatal lethality with a significantly decreased amino acid pool[13]. This suggests that RIPK1 may serve as a regulator that is responsible for starvation resistance at birth and supports neonatal survival.

Metabolic alteration plays a pivotal role in the response to starvation, which enables cell survival and maintains organismal function[14–16]. Given that metabolic balance is critical to starvation resistance, how cells sense starvation and reprogram metabolism is still elusive. RIPK1 has been reported as a sensor of various stresses[17,18]. Emerging evidence has delineated the regulatory role of RIPK1 in metabolism. For example, the loss of RIPK1 in lung cancer cells impairs mitochondrial oxidative phosphorylation and accelerates glycolysis[19]. Thus, we hypothesized that RIPK1 may also play a vital role in starvation resistance by regulating cellular metabolism.

In this study, we first discover that RIPK1 is required for cell survival under starvation conditions but not under normal conditions. Then, we employ metabolomics to profile the metabolic changes responding to $Ripk1$ deficiency and discover the dysregulation of aspartate metabolism in $Ripk1$-deficient conditions. Amino acids play an important role in regulating various cellular processes, including autophagy[20]. For example, the concentrations of amino acids in neonatal $Atg5^{-/-}$ mice are significantly lower than those in wild-type (WT) mice[13]. In particular, aspartate has been reported as an essential metabolite for cell growth[21,22]. Then, through combined metabolomics, RNA-sequencing (RNA-seq), and mechanistic studies, we reveal that $Ripk1$ deficiency modulates aspartate catabolism and disrupts metabolic homeostasis, which inactivates starvation-induced autophagy. The results demonstrate that RIPK1 is a nutrient stress sensor and metabolic regulator that is critical to maintaining metabolic homeostasis under starvation conditions.

## Results

### RIPK1 is essential for survival under starvation conditions by activating autophagy.
$Ripk1$ deficiency had little effect on cell survival under normal or glucose starvation conditions but significantly inhibited cell survival under the deprivation of total nutrients (Earle's balanced salt solution, EBSS; Fig. 1a, b and Supplementary Fig. 1a). Under starvation, cells activate catabolism (e.g., autophagy) and suppress anabolism (e.g., protein synthesis). We further inhibited protein synthesis using cycloheximide (CHX). However, under CHX treatment and EBSS starvation conditions, $Ripk1$ deficiency still inhibited cell survival significantly (Fig. 1a, b). The results suggest that RIPK1 may regulate catabolism instead of anabolism. In accordance, $Ripk1^{-/-}$ mouse neonates died in a short time without milk feeding after delivery

(Fig. 1c), and the survival time was slightly prolonged with milk feeding (Fig. 1d), suggesting that $Ripk1^{-/-}$ neonates may die due to defects in starvation resistance. We further measured the autophagy levels in tissues obtained from starved WT, $Ripk1^{-/-}$ and $Ripk1^{+/-}$ mouse neonates. Transmission electron microscopy (TEM) imaging showed the abundant formation of autophagic vacuoles, including autophagosomes (AP) and autolysosomes (AL), in the heart tissues from starved WT neonates. But these vacuoles were largely absent in $Ripk1^{-/-}$ neonates (Fig. 1e, f). Immunofluorescent staining of LC3 dots in the heart tissues further demonstrated that the formation of autophagosomes was significantly upregulated at 6 h after delivery in WT neonates (Fig. 1g, h). In contrast, significantly fewer LC3 puncta were observed in starved $Ripk1^{-/-}$ neonates. Similar results were observed in the mouse brain tissues (Supplementary Fig. 1b–e). Western blot (WB) results further validated that autophagy was induced by postnatal starvation in WT and $Ripk1^{+/-}$ neonates, as indicated by the upregulation of LC3-II protein levels and degradation of SQSTM1/p62 (p62) protein levels, but inhibited in $Ripk1^{-/-}$ neonates (Fig. 1i).

To investigate the regulatory role of RIPK1 in autophagy, mouse embryonic fibroblast (MEF) cells with $Ripk1$ knockout were used. WT and $Ripk1^{-/-}$ MEFs were incubated in EBSS to mimic nutritional deficiency conditions. Interestingly, $Ripk1$ deficiency promoted autophagy under normal conditions (culture medium) but inhibited autophagy under starvation conditions (EBSS, Fig. 1j). This result is similar to the phenomenon that $Ripk1$ deficiency promotes survival in the prenatal period (sufficient nutrient supply) but inhibits survival in the neonatal period (interrupted nutrient supply)[7–12]. Autophagy is important for survival in neonatal starvation, further suggesting that RIPK1 plays a role in starvation resistance. We found that autophagy was induced in WT cells in a time-dependent manner but not in $Ripk1^{-/-}$ cells, as indicated by the upregulation of LC3-II protein levels and downregulation of p62 protein levels (Fig. 1k). The downregulated levels of autophagy were successfully rescued by overexpressing $Ripk1$ in $Ripk1^{-/-}$ cells (i.e., $Ripk1^{-/-} + Ripk1$ cells, Fig. 1k). Furthermore, we used a stable green fluorescent protein (GFP)-LC3-transfected H4 cell line (i.e., H4-GFP-LC3)[23] and transiently transfected the cells with siRNAs targeting $Ripk1$. As shown in Fig. 1l, m, GFP-LC3 puncta were not observed under starvation conditions in the $Ripk1$ knockdown groups, suggesting that autophagosome formation was inhibited by $Ripk1$ deficiency. Taken together, these results demonstrated that RIPK1 is essential for starvation resistance under starvation conditions.

### Aspartate is a RIPK1-dependent metabolite under starvation.
We first profiled the metabolic changes responding to $Ripk1$ deficiency in both MEFs and Jurkat cells under normal conditions using liquid chromatography–mass spectrometry (LC–MS)-based targeted metabolomics[24]. A total of ~200 essential metabolites were measured, and 68 metabolites were significantly changed in $Ripk1^{-/-}$ MEFs (Student's $t$-test, $p < 0.05$; and fold change >1.2; Fig. 2a). Metabolic pathway enrichment analysis further revealed that the alanine, aspartate, and glutamate metabolism pathway was one of the enriched pathways (Fig. 2b, c). Similar results were also obtained using $RIPK1^{-/-}$ Jurkat cells (Supplementary Fig. 2a–c). Then, under starvation conditions, we confirmed that aspartate levels were consistently upregulated in $Ripk1^{-/-}$ cells and rescued by overexpressing $Ripk1$ in $Ripk1^{-/-}$ cells (i.e., $Ripk1^{-/-} + Ripk1$ cells, Fig. 2d, e). Similar results were also validated in the early neonatal $Ripk1^{-/-}$ mouse tissues (Fig. 2f, g and Supplementary Fig. 2d, e). By checking all RIPK1-dependent metabolites from the mouse brain tissue, liver tissues, MEFs, and

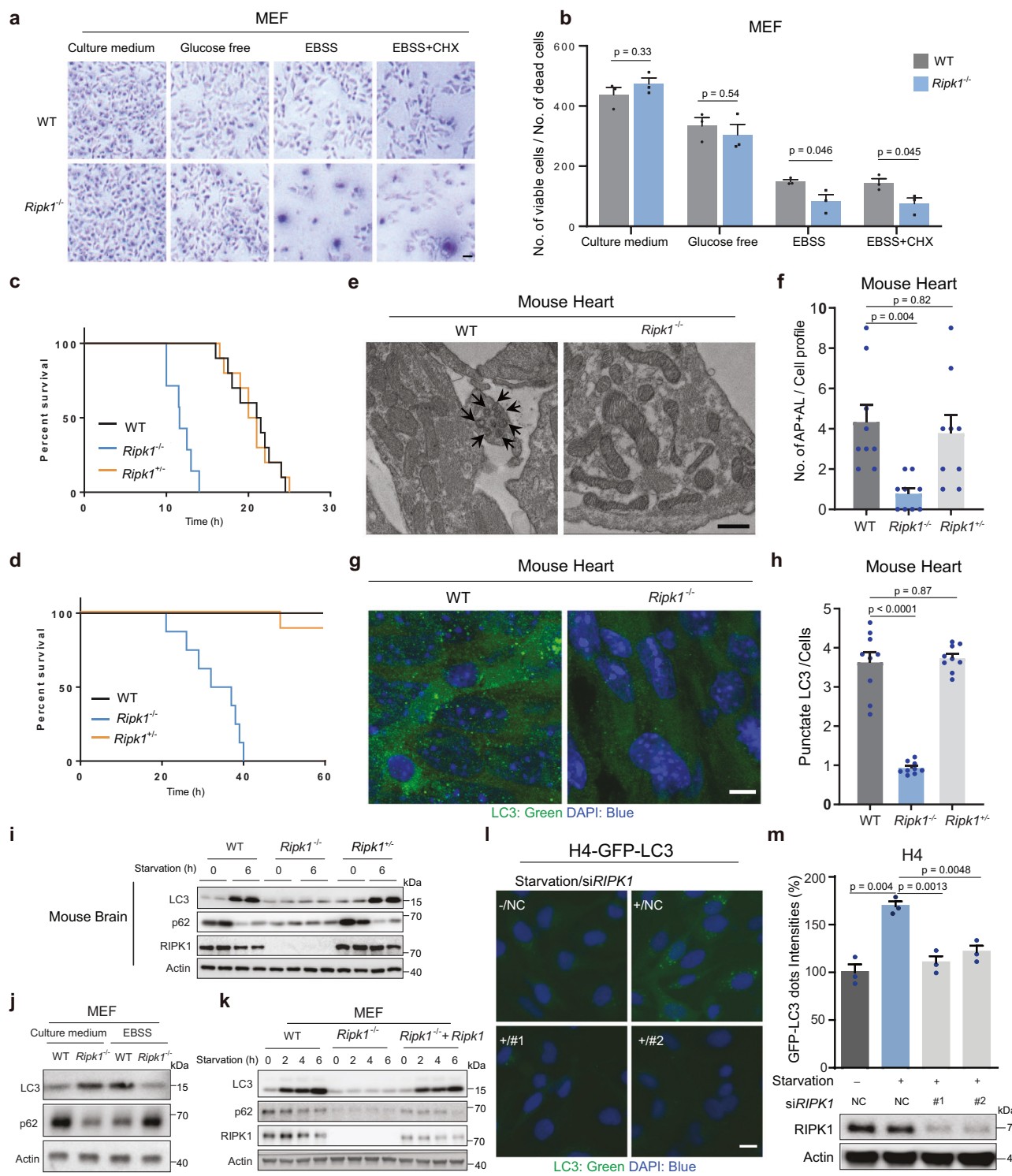

Jurkat cells under starvation conditions (Fig. 2d–g), we found that aspartate was the specific metabolite that consistently responded to RIPK1 expression in all four biological sample groups under starvation conditions. These results indicate that RIPK1 deficiency causes increased aspartate levels in cells and mouse tissues, which are rescued by RIPK1 complementation.

**Increased aspartate induced by RIPK1 deficiency inhibits starvation-induced autophagy.** We further studied the effects of intracellular aspartate levels on starvation-induced autophagy.

Under starvation, increased aspartate levels by adding aspartate to WT MEFs led to decreased LC3-II and increased p62 levels in a dose-dependent manner (Fig. 3a, b). Consistently, in starved H4-GFP-LC3 cells, aspartate treatment significantly decreased the formation of autophagosomes (Fig. 3c, d). In contrast, we also reduced the intracellular aspartate levels in $Ripk1^{-/-}$ MEFs by knocking down glutamic-oxaloacetic transaminase 2 (GOT2), which is a key enzyme involved in intracellular synthesis of aspartate (Fig. 3e). As expected, the knockdown of $Got2$ dramatically restored autophagy levels in $Ripk1^{-/-}$ MEFs under starvation conditions. LC3-II protein levels were increased while p62

**Fig. 1 RIPK1 is essential for survival under starvation conditions by activating autophagy. a, b** Assessments of cell survival by trypan blue exclusion. WT and *Ripk1*−/− MEFs were cultured in culture medium (6 h), glucose-free medium (6 h), EBSS (6 h), or EBSS with CHX (6 h), respectively. Scale bar represents 10 μm. Bars in (**b**) represent mean ± SEM (*n* = 3 biologically independent samples). *P* values were determined by a two-tailed Student's *t*-test. **c, d** Mouse pup survival for different *Ripk1* genotypes. Neonates were monitored in chambers without milk feeding (**c**) or with milk feeding (**d**). Bars represent mean ± SEM (c, WT, n = 10; Ripk1−/−, n = 7; Ripk1+/−, n = 10; d, WT, n = 10; Ripk1−/−, n = 8; Ripk1+/−, *n* = 10; all numbers are biologically independent samples). **e** Representative TEM images of the heart tissues obtained from WT and *Ripk1*−/− mouse neonates at 6 h after birth. The arrows indicated the formed autophagic vacuoles. Scale bar represents 500 nm. **f** The numbers of autophagic vacuoles (AP autophagosome, AL autolysosome) in TEM images of heart tissues collected from WT, *Ripk1*−/− and *Ripk1*+/− mouse neonates 6 h after birth. Bar results represent mean ± SEM obtained from nine random cell profiles in each electron microscopy section per experimental group (three mice per group, three images per mouse). *P* values were determined by a two-tailed Student's *t*-test. **g, h** Immunofluorescence (IF) staining of mouse heart tissues for LC3 dot intensities (green) and DAPI (blue) from WT and *Ripk1*−/−mouse neonates at 6 h after birth without milk feeding. Representative IF images were shown in (**g**). Scale bar represents 10 μm. The fluorescent intensities of LC3 dots/cells were quantified in (**h**) using ImageJ. Bars represent mean ± SEM (*n* = 9 images from three mice per group). *P* values were determined by a two-tailed Student's *t*-test. **i** Western blot analyses of LC3 and p62 levels in brain tissues from WT, *Ripk1*−/− and *Ripk1*+/− mouse neonates at 0 or 6 h after birth. **j** Western blot analyses of LC3 and p62 levels in WT and *Ripk1*−/− MEFs which were cultured in culture medium or EBSS for 6 h. **k** Western blot analyses of LC3 and p62 levels in WT, *Ripk1*−/− and *Ripk1*−/− + *Ripk1* MEFs which were starved for 0, 2, 4, or 6 h in EBSS. **l** Representative fluorescent images of punctate GFP-LC3 responding to *RIPK1* knockdown. H4-GFP-LC3 cells were transfected with nontarget control (NC) or *RIPK1* siRNAs (#1 and #2). Cells were cultured in normal or starvation conditions (EBSS) for 4 h before harvest. Cells were stained with DAPI and visualized by microscopy. Scale bar represents 10 μm. **m** The averaged punctate GFP-LC3 intensities in images from each condition in (**l**). The fluorescent intensities were normalized to the "starvation-/NC" group. Bars represent mean ± SEM (*n* = 3 biologically independent samples per group, each point represents the mean intensity of three images in each sample). *P* values were determined by a two-tailed Student's *t*-test. The immunoblot analyses showed the knockdown efficiencies of siRNAs targeting *RIPK1*.

protein levels were decreased in response to *Got2* knockdown (Fig. 3f). Among the shRNAs targeting GOT2, #1 shRNA induced the most significant decrease of GOT2 level (Fig. 3f). Accordingly, we consistently observed the most significant decrease of aspartate level and increase of autophagy in *Ripk1*−/− cells with #1 shRNA (Fig. 3e), suggesting that aspartate inhibiting autophagy induction under starvation is proportional to the concentrations of aspartate. The results were further validated through a GFP-LC3 assay in *RIPK1*−/− HEK293T cells (Fig. 3g, h and Supplementary Fig. 3a). Similar results were obtained with the treatment of a GOT2 inhibitor, aminooxy acetic acid (AOA)[25]. The inhibition of GOT2 decreased the intracellular aspartate levels in *Ripk1*−/− MEFs (Fig. 3i) and activated autophagy, as shown by the increased levels of LC3-II, reduced levels of p62 (Fig. 3j), and increased numbers of GFP-LC3 puncta in H4-GFP-LC3 cells with *RIPK1* knockdown (Fig. 3k–m). Although AOA had a massive effect on aspartate (Fig. 3i), it generated a very similar effect on autophagy induction, which may be due to the side effect of AOA treatment. AOA was reported as an inhibitor of GOT2 as well as an inhibitor of cystathionine-beta-synthase (CBS)[26]. In addition, we used relatively high concentrations of AOA for treatment, which may have toxic effects on cells. Taken together, these results suggest that the increased aspartate levels in *Ripk1*−/− cells underlie the mechanism of defective starvation-induced autophagy. In addition, we measured mRNA expression in WT and *Ripk1*−/− MEFs using RNA-seq. Among the 3,160 significantly changed genes in *Ripk1*−/− MEFs (*p* < 0.05; Student's *t*-test), pathway enrichment showed that autophagy-related genes were not enriched (Supplementary Fig. 3b). Individual gene expression analysis also showed that the autophagy-related genes had no significant changes in the *Ripk1*−/− cells (Supplementary Fig. 3c). These results demonstrated that RIPK1 regulated starvation-induced autophagy by modulating metabolism rather than autophagy-related genes.

**Increased aspartate inhibits the AMPK pathway in RIPK1-deficient cells under starvation.** Previous reports have revealed the regulatory role of AMPK in autophagy in response to nutrient starvation[27,28]. AMPK senses changes in intracellular AMP/ATP ratio to maintain energy homeostasis[29]. Our data first showed that AMP/ATP ratios were decreased in both newborn *Ripk1*−/− mice and cultured cells (Fig. 4a, b and Supplementary Fig. 4a, b). Consistently, we found that the activation of AMPK through AMPK phosphorylation (AMPKα Thr172) and phosphorylation of its downstream substrate acetyl-CoA carboxylase (ACC Ser79) was blocked by *Ripk1* deficiency in both neonatal mouse brains and EBSS-cultured MEFs (Fig. 4c, d and Supplementary Fig. 4c–e). These observations were consistent with the low autophagy levels in starved *Ripk1*−/− mice and *Ripk1*−/− MEFs (Fig. 1). Previous reports have demonstrated that AMPK regulates autophagy activation through the direct phosphorylation of ULK1 (mammalian homolog of yeast Atg1)[27,30]. Here, we confirmed that phosphorylation of ULK1 (Ser317) was inhibited in starved *Ripk1*−/− mice and *Ripk1*−/− MEFs (Supplementary Fig. 4d, e). In contrast, when *Ripk1*−/− cells were treated with aminoimidazole carboxamide ribonucleotide (AICAR), a direct activator of AMPK, both autophagy and p-AMPK/p-ULK1 levels were upregulated under starvation (Fig. 4e). Thus, the defective autophagy induction in *Ripk1*−/− mice and cells was ascribed to the defective activation of the AMPK signaling pathway.

We further demonstrated that the increased aspartate levels reduced the AMP/ATP ratios in both starved WT Jurkat cells and MEFs (Fig. 4f). Consistently, the starvation-induced elevation of p-AMPK, p-ACC, and p-ULK1 levels in WT cells was reduced upon aspartate treatment (Fig. 4g). The defective activation of AMPK in starved WT MEFs inhibited by aspartate treatment was further restored by the direct activation of AMPK using AICAR (Fig. 4h). In comparison, the reduction in intracellular aspartate levels in *Ripk1*−/− cells through *Got2* knockdown significantly increased the AMP/ATP ratios (Fig. 4i). As a result, p-AMPK and p-ULK1 levels were increased in response to GOT2 knockdown (Fig. 4j). Similar results were observed when GOT2 activities were inhibited by AOA treatment (Fig. 4k, l). Altogether, increased intracellular aspartate regulated autophagy by inhibiting the AMPK pathway.

**RIPK1 deficiency increases ATP production by enhancing TCA flux.** We demonstrated that the increased aspartate levels in *Ripk1*−/− cells decreased the AMP/ATP ratios and inhibited AMPK-mediated autophagy. Thus, energy metabolism in *Ripk1*−/− cells should be significantly altered. We performed cell

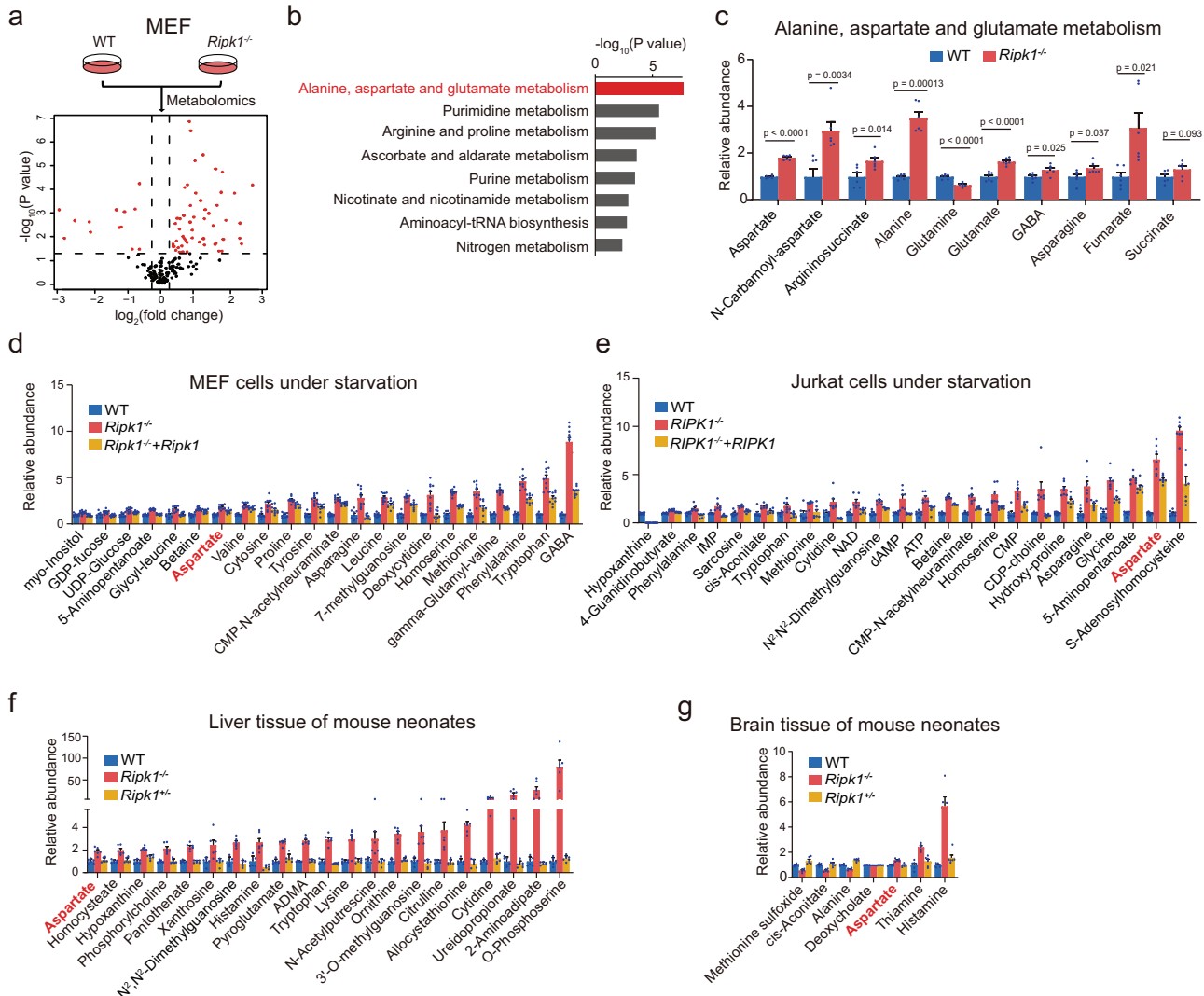

**Fig. 2 Aspartate is a RIPK1-dependent metabolite under starvation. a** Volcano plot of metabolites in WT and $Ripk1^{-/-}$ MEFs under normal culture conditions. $P$ values were determined by a two-tailed Student's $t$-test. Red dots indicate significantly changed metabolites ($p < 0.05$ and fold change >1.2, $n = 6$ biologically independent samples in each group). **b** Pathway enrichment analysis of significantly changed metabolites in MEFs in (**a**). **c** Relative abundances of metabolites in the alanine, aspartate, and glutamate metabolism pathway were measured in MEFs. $n = 6$ biologically independent samples in each group. $P$ values were determined by a two-tailed Student's $t$-test. **d, e** Metabolites were significantly increased/decreased in $Ripk1^{-/-}$ group compared with WT group ($p < 0.05$), and significantly rescued ($p < 0.05$) in $Ripk1^{-/-} + Ripk1$ group compared with $Ripk1^{-/-}$ group: MEFs (**d**, WT group, $n = 9$; $Ripk1^{-/-}$ group, $n = 10$; $Ripk1^{-/-} + Ripk1$ group, $n = 10$); Jurkat cells (**e**, $n = 7$). All cells were cultured in EBSS for 4 h. All numbers are biologically independent samples. $P$ values were determined by a two-tailed Student's $t$-test with FDR correction. **f, g** Metabolites were significantly ($p < 0.05$) increased/decreased in $Ripk1^{-/-}$ compared with WT, and significantly rescued ($p < 0.05$) in $Ripk1^{+/-}$ compared with $Ripk1^{-/-}$ in mouse liver (**f**, $n = 6$ biologically independent samples) and brain (**g**, WT group, $n = 6$; $Ripk1^{-/-}$ group, $n = 5$; $Ripk1^{-/-} + Ripk1$ group, $n = 6$) under starvation. All numbers are biologically independent samples. $P$ values were determined by a two-tailed Student's $t$-test with FDR correction. Bar graphs represent mean ± SEM.

energy metabolism analyses of WT and $Ripk1^{-/-}$ MEFs using Seahorse. Specifically, measurements of oxygen consumption rates (OCRs) demonstrated that ATP production was upregulated in starved $Ripk1^{-/-}$ MEFs (Fig. 5a). Similar results were obtained in starved WT MEFs treated with aspartate (375 μM, Fig. 5b). The elevation of ATP levels in $Ripk1^{-/-}$ cells and aspartate-treated WT cells was further confirmed by LC–MS analysis (Fig. 5c, d). Although aspartate treatment has been reported to increase ATP production by promoting glycolysis in tumor cells[31], our data showed that ECAR values had no significant changes in $Ripk1^{-/-}$ MEFs and aspartate-treated WT MEFs (Fig. 5e, f and Supplementary Fig. 5a, b). The increased production of ATP may be ascribed to the enhanced activity of TCA cycle. Indeed, LC–MS

analyses demonstrated that metabolites in TCA cycle, such as citrate and $cis$-aconitate, were elevated in $Ripk1^{-/-}$ MEFs and Jurkat cells under starvation (Fig. 5g and Supplementary Fig. 5c). Similarly, aspartate treatment also increased TCA metabolites in MEFs and Jurkat cells (Fig. 5h and Supplementary Fig. 5d). Consistently, glycolysis metabolites showed no changes in response to RIPK1 deficiency and aspartate treatment (Supplementary Fig. 5e, f). We further employed stable-isotope tracing technology with [U-$^{13}$C]-aspartate to validate the enhanced TCA activity in response to $Ripk1$ deficiency. The results in starved WT and $Ripk1^{-/-}$ cells clearly demonstrated that aspartate-derived metabolites in TCA cycle, such as citrate, $cis$-aconitate, α-keto-glutarate, and malate, were significantly increased in $Ripk1^{-/-}$

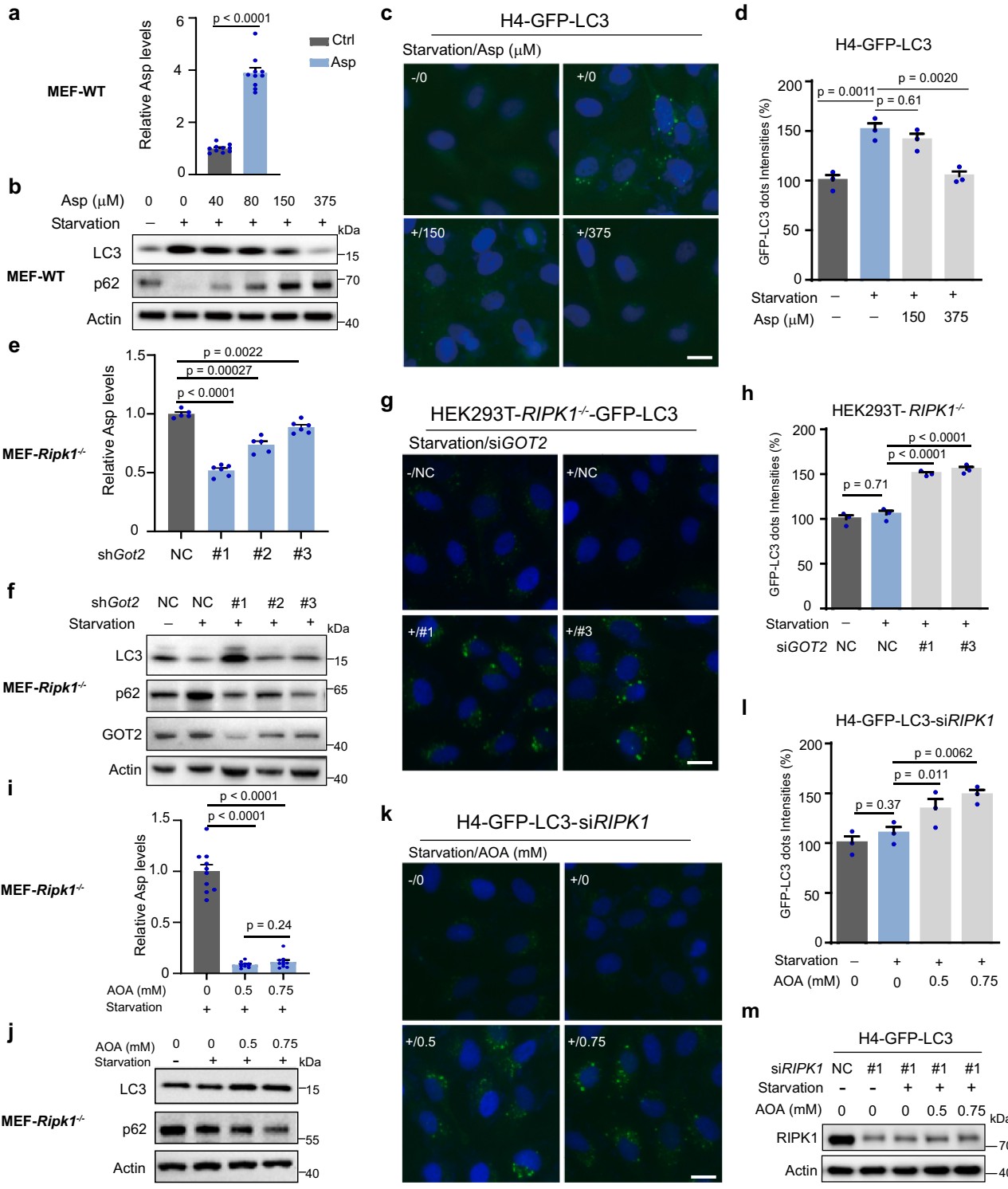

cells (Fig. 5i, j). Similarly, with the treatment of [U-$^{13}$C]-gluta-mine, the fully labeled TCA metabolites and aspartate were also significantly higher in $Ripk1^{-/-}$ MEFs (Supplementary Fig. 5g, h). These results suggest that the increased aspartate level in $Ripk1^{-/-}$ cells contributes to the increased ATP production by enhancing the activity of TCA cycle, which eventually causes energy metabolic disorder under starvation.

**RIPK1 deficiency suppresses aspartate catabolism by inactivating SP1.** To understand how $Ripk1$ deficiency affects aspartate metabolism, we checked the $Ripk1$-dependent genes associated with aspartate metabolism. RNA-seq data showed that the expressions of the aspartate/glutamate transporter $Slc1a1$ (Solute carrier family 1 member 1) and $Slc1a3$ were decreased in $Ripk1^{-/-}$ MEFs (Fig. 6a and Supplementary Fig. 6a). The results suggestd

**Fig. 3 Increased aspartate induced by RIPK1 deficiency inhibits starvation-induced autophagy. a** Intracellular aspartate levels in WT MEFs under starvation (EBSS, 4 h) responding to the aspartate treatment were determined by LC-MS (Ctrl, 0; Asp, 375 μM of Asp; $n = 10$ biologically independent samples per group). $P$ values were determined by a two-tailed Student's $t$-test. **b** Western blot analyses of LC3 and p62 levels in WT MEFs which were incubated under starvation (EBSS) with the aspartate supplements for 4 h. **c** Representative images of autophagosome formation in H4-GFP-LC3 cells responding to the aspartate treatments. Cells were treated with aspartate for 4 h under normal or starvation conditions (EBSS). Images were captured using microscopy. Scale bar represents 10 μm. **d** The average punctate GFP-LC3 intensities in images from each condition in (**c**). The fluorescence intensities were measured using an ArrayScan HCS 4.0 reader. Data were normalized to the starvation-/aspartate-untreated group. $n = 3$ biologically independent samples per group, each point represents the mean intensity of three images in each sample. $P$ values were determined by one-way ANOVA by Tukey's multiple comparisons test. **e** Intracellular aspartate levels in $Ripk1^{-/-}$ MEFs were measured using LC-MS. $Ripk1^{-/-}$ MEFs were transfected with nontarget control (NC) or *Got2* shRNAs (#1–3), and cultured in EBSS for 4 h (NC, $n = 5$; #1, $n = 6$; #2, $n = 5$; #3, $n = 6$. All numbers are biologically independent samples). $P$ values were determined by a two-tailed Student's $t$-test. **f** Western blot analyses of LC3, p62, and GOT2 in $Ripk1^{-/-}$ MEFs. **g** Representative images of autophagosome formation in $RIPK1^{-/-}$ HEK293T cells which were transfected with GFP-LC3 plasmids and then NC or GOT2 siRNAs (#1, #3). Cells were stained with DAPI and captured by microscopy. Scale bar represents 10 μm. **h** The average intensity of LC3 dots from each indicated sample in (**g**). Fluorescent intensities were measured using the ArrayScan HCS 4.0 reader. Data were normalized to the starvation-/NC group. $n = 3$ biologically independent samples per group, each point represents the mean intensity of three images in each sample. $P$ values were determined by one-way ANOVA by Tukey's multiple comparisons test. **i** Intracellular aspartate levels in $Ripk1^{-/-}$ MEFs responding to the AOA treatment (0, 0.5, and 0.75 mM) in EBSS for 4 h were determined by LC–MS ($n = 10$ biologically independent samples per group). $P$ values were determined by a two-tailed Student's $t$-test. **j** Western blot analyses of LC3 and p62 levels in $Ripk1^{-/-}$ MEFs responding to the AOA treatment in EBSS for 4 h as indicated. **k** Representative images were captured by microscopy in H4-GFP-LC3 cells which were transfected with $RIPK1$ siRNAs and then treated in the absence or presence of AOA for 4 h in culture medium or EBSS as indicated. Scale bar represents 10 μm. **l** The average intensity of LC3 dots from each indicated sample in (**k**). The fluorescence intensities were measured by the ArrayScan HCS 4.0 reader. Data were normalized to the starvation-/AOA-untreated group. $n = 3$ biologically independent samples per group, each point represents the mean intensity of three images in each sample. $P$ values were determined by one-way ANOVA by uncorrected Fisher's LSD test. **m** Western blot analyses of RIPK1 and Actin showed the knockdown efficiency of RIPK1 in (**k**) and (**l**). Bar graphs represent mean ± SEM.

that the increased intracellular aspartate in $Ripk1^{-/-}$ cells is not from extracellular transportation. This was further confirmed by the measurements of intracellular aspartate levels in aspartate-treated WT and $Ripk1^{-/-}$ cells (Supplementary Fig. 6b). We also found that genes encoding aspartate catabolic enzymes, including *Got1* (glutamate oxaloacetate transaminase 1), *Asns* (asparagine synthetase), *Ass1* (argininosuccinate synthetase 1), *Adss* (adenylosuccinate synthetase), and *Cad* (aspartate transcarbamylase), were downregulated in $Ripk1^{-/-}$ MEFs (Fig. 6a and Supplementary Fig. 6c–e). The results were confirmed by real-time PCR analyses in MEFs (Fig. 6b). Meanwhile, in the [U-$^{13}$C]-aspartate-based isotope tracing experiment, we found that the degradation metabolites of aspartate, such as adenylosuccinate and orotate, were significantly decreased in $Ripk1^{-/-}$ cells under starvation conditions (Fig. 6c). Therefore, RIPK1 may increase intracellular aspartate levels by inhibiting aspartate catabolism. Transcription factor prediction analysis showed that genes encoding aspartate catabolic enzymes share a GC-rich motif in the [−2 kb, 2 kb] region, which can be recognized by the transcription factor SP1 (Fig. 6d and Supplementary Table 1 and 2). SP1 has been reported to activate the promoter activity of *Got1*[32], *Cad*[32,33], *Asns*[34], and *Ass1*[35]. Real-time PCR analysis validated that *Sp1* knockdown decreased the expressions of *Got1*, *Ass1*, *Asns*, *Adss*, and *Cad* under starvation (Fig. 6e). These results suggest that RIPK1 regulates SP1 to further influence the expressions of aspartate catabolism genes. Although the mRNA and protein expression levels of SP1 were not affected by RIPK1 deficiency (Supplementary Fig. 6f, g), the luciferase activities of SP1 were significantly downregulated in $Ripk1^{-/-}$ cells under starvation (Fig. 6f). Consistently, luciferase data showed that the promoter activities of *GOT1*, *ASS1*, *ASNS*, *ADSS*, and *CAD* were also decreased in $RIPK1^{-/-}$ HEK293T cells and reversed by SP1 overexpression under starvation (Fig. 6g).

To validate that the increased aspartate levels in $Ripk1^{-/-}$ cells because of the SP1 inactivation, intracellular aspartate levels responding to *Sp1* knockdown were determined. LC–MS data showed that aspartate levels were increased in MEFs transfected with siRNAs targeting *Sp1* (Fig. 6h) and accompanied by

decreased AMP/ATP ratios (Fig. 6i). In addition, immunoblot and GFP-LC3 assays showed that *Sp1* knockdown also inhibited autophagy levels and AMPK signaling (Fig. 6j and Supplementary Fig. 6h, i), similar to *Ripk1* deficiency. Furthermore, over-expression of SP1 abrogated *Ripk1* deficiency-induced autophagy retardation (Fig. 6k). These results suggest that RIPK1 regulates autophagy by modulating aspartate catabolism, which is achieved by negatively regulating transcription factor SP1.

**RIPK1 deficiency inactivates SP1 by inhibiting SP1 nuclear translocation.** Since RIPK1 does not affect the expression of SP1 but rather its activity, we then explored how SP1 inactivation was achieved upon *Ripk1* deficiency. The transcriptional activity of SP1 could be affected by many factors, such as binding affinity to promoters, nuclear translocation, protein stability, and interaction with other protein factors[36]. SP1 nuclear translocation was increased under starvation conditions (Fig. 7a). In WT MEFs, we found that the SP1 expression was largely increased in the nucleus, while nuclear SP1 was significantly decreased in $Ripk1^{-/-}$ MEFs (Fig. 7b, c). SP1 nuclear translocation can be regulated by its phosphorylation[36,37]. Accordingly, we also found that phosphorylated SP1 (T739) was significantly decreased in $Ripk1^{-/-}$ MEFs (Fig. 7d). To evaluate whether RIPK1 kinase activity is required for SP1 nuclear translocation, necrostain-1 (Nec-1), a RIPK1 kinase activity inhibitor, was used[38]. Nuclear separation data showed that Nec-1 did not affect SP1 nuclear translocation under starvation and normal conditions (Fig. 7e and Supplementary Fig. 7b). Our IHC data confirmed that Nec-1 had little effect on SP1 nuclear translocation under starvation conditions (Supplementary Fig. 7a). We also found that Nec-1 treatment had no effect on SP1 phosphorylation and autophagy activation under various conditions (Fig. 7f). Transfection of the RIPK1 kinase-dead mutant (K45M) rescued SP1 nuclear localization in $Ripk1^{-/-}$ MEFs, suggesting the kinase-independent regulation of RIPK1 (Fig. 7g). Together, the results suggest that RIPK1 regulates SP1 nuclear expression and phosphorylation in a kinase-independent manner. Although our data indicated that RIPK1 and SP1

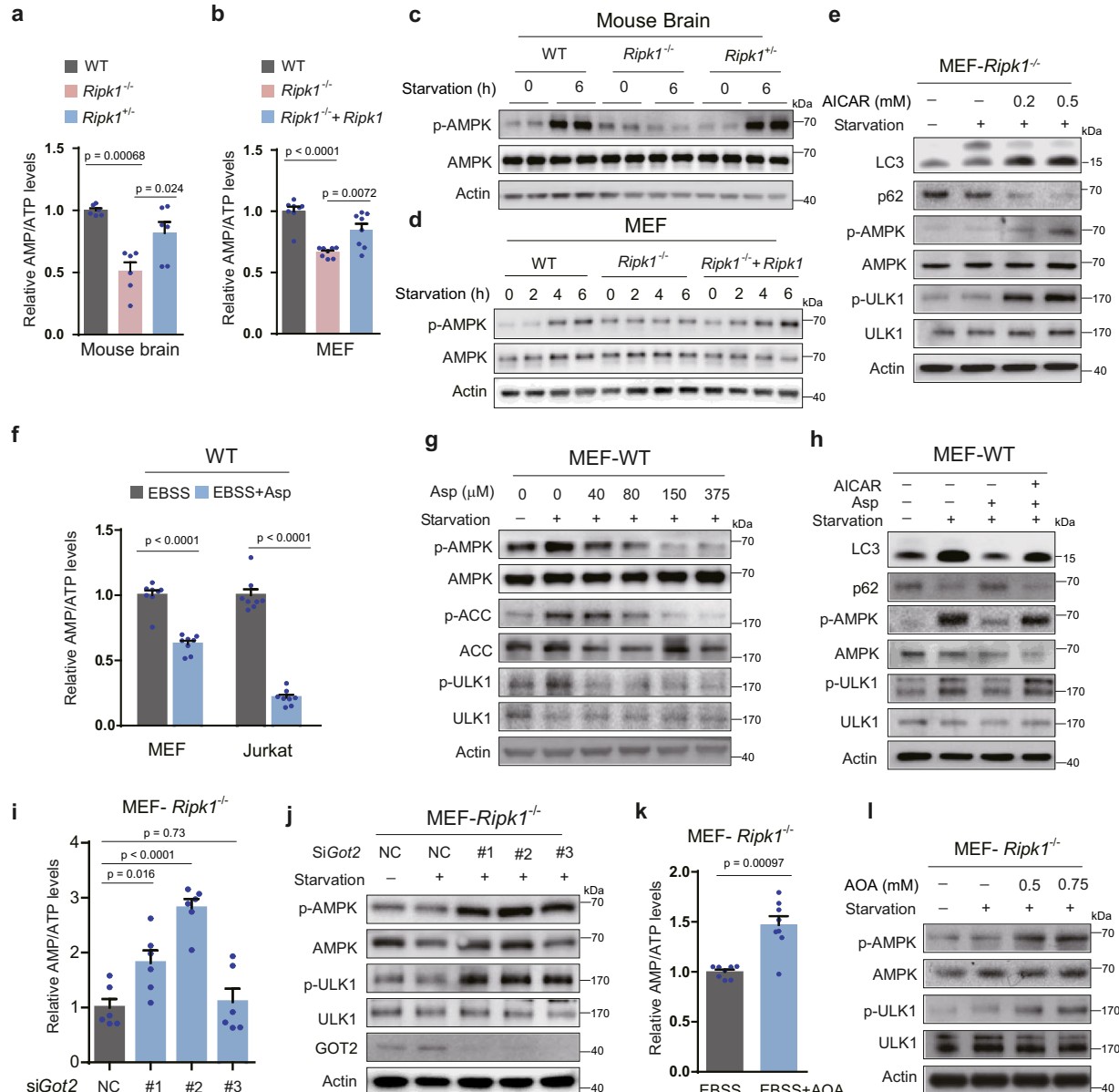

**Fig. 4 Increased aspartate inhibits the AMPK pathway in RIPK1-deficient cells under starvation. a** Relative AMP/ATP ratios in brain tissues obtained from WT, $Ripk1^{-/-}$, and $Ripk1^{+/-}$ mouse neonates were measured using LC–MS. The ratios were normalized to the WT group ($n = 6$ biologically independent samples per group). $P$ values were determined by a two-tailed Student's $t$-test. **b** Relative intracellular AMP/ATP ratios in WT, $Ripk1^{-/-}$, and $Ripk1^{-/-} + Ripk1$ MEFs by the luminescent assay ($n = 8$ biologically independent samples per group). All cells were cultured in EBSS for 4 h before collection. $P$ values were determined by a two-tailed Student's $t$-test. **c** Western blot analyses of p-AMPK and AMPK levels in mouse neonate brain. Neonate brain tissues were collected at 0 or 6 h after birth. **d** Western blot analyses of p-AMPK and AMPK levels in WT, $Ripk1^{-/-}$, and $Ripk1^{-/-} + Ripk1$ MEFs. Cells were cultured under EBSS conditions for 0, 2, 4, or 6 h. **e** Western blot analyses of $Ripk1^{-/-}$ MEFs which were treated with 0, 0.2, and 0.5 mM AICAR in culture medium or EBSS for 4 h. **f** Relative intracellular AMP/ATP ratios in WT MEFs and Jurkat cells were measured by the luminescent assay. Cells were cultured under EBSS or EBSS supplemented with aspartate (375 μM) for 4 h before harvest ($n = 8$ biologically independent samples per group). $P$ values were determined by a two-tailed Student's $t$-test. **g** Western blot analyses of p-AMPK, AMPK, p-ACC, ACC, p-ULK1, and ULK1 levels in WT MEFs treated with aspartate as indicated. **h** Western blot analyses of LC3, p62, p-AMPK, AMPK, p-ULK1, and ULK1 levels in WT MEFs treated with AICAR (0.5 mM, 4 h) and aspartate (0.15 mM; 4 h) in EBSS as indicated. **i** Relative intracellular AMP/ATP ratios in $Ripk1^{-/-}$ MEFs were detected by the luminescent assay. $Ripk1^{-/-}$ MEFs were transfected with nontarget control (NC) or $Got2$ siRNAs (#1–3), and cultured in EBSS for 4 h ($n = 6$ biologically independent samples per group). $P$ values were determined by a two-tailed Student's $t$-test. **j** Western blot analyses of p-AMPK, AMPK, p-ULK1, and ULK1 levels in $Ripk1^{-/-}$ MEFs responding to $Got2$ knockdown. $Got2$ knockdown efficiency was detected using GOT2 antibodies. Cells were transfected with NC or $Got2$ siRNAs (#1–3), and cultured in normal medium or EBSS for 4 h. **k** Relative intracellular AMP/ATP ratios in $Ripk1^{-/-}$ MEFs responding to AOA treatment (0.75 mM, 4 h) under starvation conditions (EBSS, 4 h). AMP and ATP were detected by the luminescent assay. The ratios were normalized to the AOA-untreated group ($n = 8$ biologically independent samples per group). $P$ values were determined by a two-tailed Student's $t$-test. **l** Western blot analyses of p-AMPK, AMPK, p-ULK1, and ULK1 levels in $Ripk1^{-/-}$ MEFs responding to AOA treatment. Cells were cultured for 4 h in culture medium or EBSS with or without AOA as indicated. Bar graphs represent mean ± SEM.

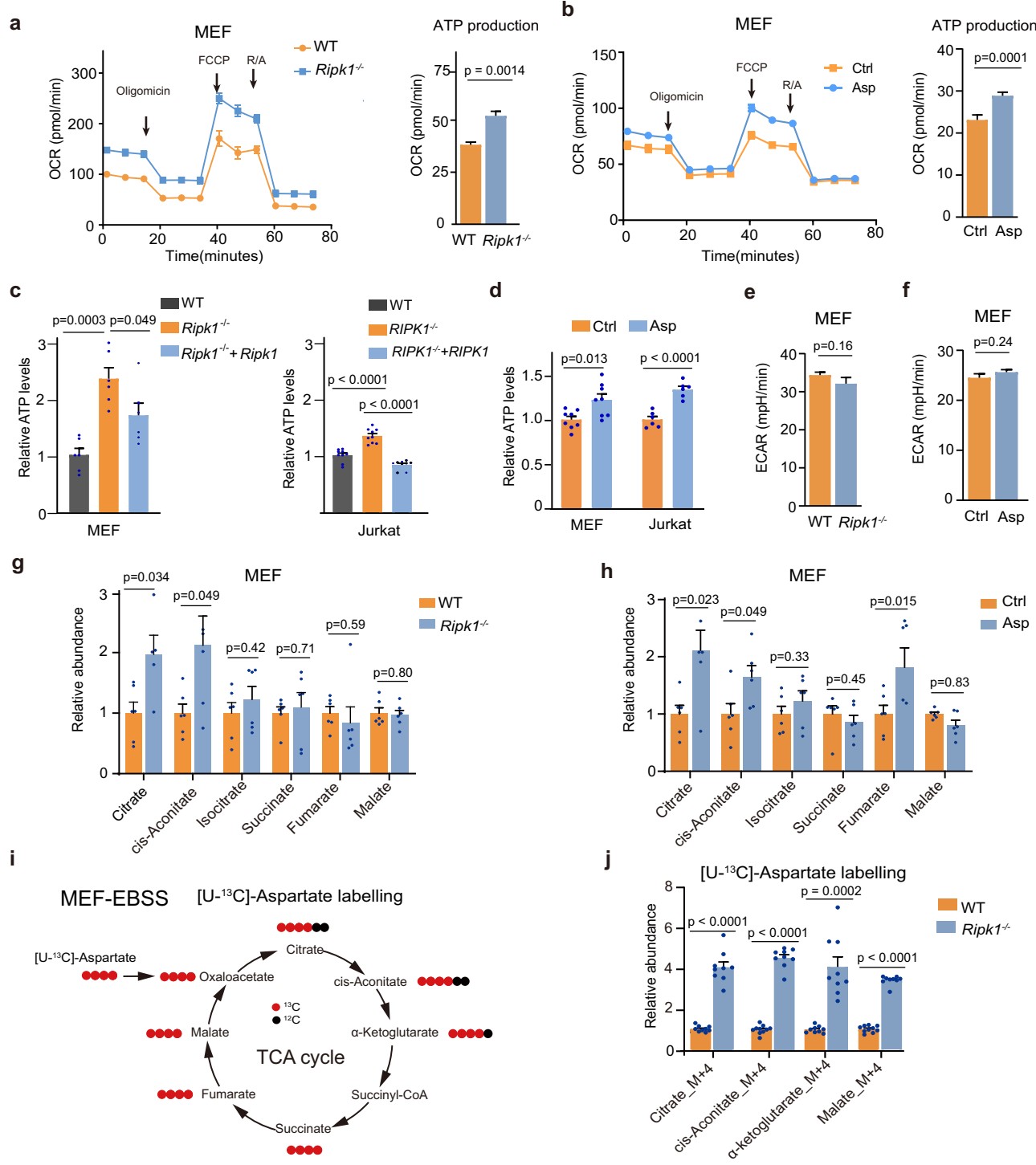

colocalized in the cytosol and nucleus (Supplementary Fig. 7c, d), and RIPK1 interacted with SP1 (Supplementary Fig. 7e), the mechanism by which RIPK1 regulates SP1 translocation and activity requires more experimental evidences.

**Reduced aspartate levels rescue the survival of *Ripk1*$^{-/-}$ cells from starvation.** To determine whether aspartate accumulation is a byproduct of the mechanism by which RIPK1 deficiency fails to protect cells from starvation, we then tested the survival of *Ripk1*$^{-/-}$ cells responding to the reduced intracellular aspartate. Our results showed that AOA treatment decreased aspartate,

increased AMP/ATP levels, and activated autophagy in *Ripk1*$^{-/-}$ cells (Figs. 3 and 4). We further demonstrated that upon AOA treatment, *Ripk1*$^{-/-}$ MEFs showed higher cell viability than untreated cells under starvation conditions (Fig. 8a, b). SP1 overexpression, which upregulated the expression of aspartate catabolism genes and decreased aspartate in *Ripk1*$^{-/-}$ cells (Fig. 8c), rescued the viability of *RIPK1*$^{-/-}$ HEK293T cells under starvation conditions (Fig. 8d, e). Taken together, the results demonstrated that aspartate catabolism represents the underlying mechanism by which RIPK1 regulates starvation resistance (Fig. 8f).

**Fig. 5 RIPK1 deficiency increases ATP production by enhancing TCA flux. a** Oxygen consumption rates (OCR) measured in WT and $Ripk1^{-/-}$ MEFs. FCCP, carbonyl cyanide-4-(trifluoromethoxy) phenylhydrazone; R/A, rotenone/antimycin A. Cells were cultured under normal growth conditions and then starved for 4 h in EBSS before harvest. $n = 4$ biologically independent samples, shown as mean ± SEM. $P$ values were determined by a two-tailed Student's $t$-test. **b** OCR measurements in WT MEFs incubated with EBSS with or without aspartate (375 μM) for 4 h. Ctrl, $n = 34$ biologically independent samples; Asp, $n = 58$ biologically independent samples, shown as mean ± SEM. $P$ values were determined by a two-tailed Student's $t$-test. **c** Intracellular ATP levels measured by LC–MS in WT, $Ripk1^{-/-}$, $Ripk1^{-/-} + Ripk1$ MEFs ($n = 6$ biologically independent samples) and WT, $RIPK1^{-/-}$, $RIPK1^{-/-} + RIPK1$ Jurkat cells cultured in EBSS for 6 h ($n = 9$ biologically independent samples.). $P$ values were determined by a two-tailed Student's $t$-test. **d** Intracellular ATP levels measured by LC–MS in WT MEFs ($n = 6$ biologically independent samples) and WT Jurkat cells ($n = 8$ biologically independent samples). The cells were cultured in EBSS or EBSS supplemented with aspartate (375 μM) for 4 h. $P$ values were determined by a two-tailed Student's $t$-test. **e** Extracellular acidification rates (ECAR) of WT ($n = 58$ biologically independent samples) and $Ripk1^{-/-}$ MEFs ($n = 34$ biologically independent samples). Cells were starved in EBSS for 4 h. $P$ values were determined by a two-tailed Student's $t$-test. **f** ECAR measurements of MEFs treated ($n = 10$ biologically independent samples) or untreated with aspartate (375 μM, $n = 12$ biologically independent samples). Cells were starved in EBSS for 4 h. $P$ values were determined by a two-tailed Student's $t$-test. **g**, **h** Relative abundances of metabolites in TCA cycle measured using LC–MS ($n = 6$ biologically independent samples per group). WT and $Ripk1^{-/-}$ MEFs were cultured in EBSS for 4 h before harvest (**g**). WT MEFs were cultured in EBSS supplemented with 0 or 375 μM aspartate for 4 h before harvest (**h**). $P$ values were determined by a two-tailed Student's $t$-test. **i** Schematic illustration of stable isotope tracing using [U$^{13}$-C]-aspartate as a tracer. Cells were cultured with 375 μM [U$^{13}$-C]-aspartate in EBSS for 7 h before collection. Black dots represent $^{12}$C and red dots represent $^{13}$C. **j** Relative abundances of metabolites in TCA cycle after [U$^{13}$-C]-aspartate treatment of WT and $Ripk1^{-/-}$ MEFs for 7 h (WT, $n = 10$ biologically independent samples; $Ripk1^{-/-}$, $n = 9$ biologically independent samples). $P$ values were determined by a two-tailed Student's $t$-test. Bar graphs represent mean ± SEM.

## Discussion

Metabolism homeostasis is important for survival under starvation conditions[39,40]. Mice deficient in autophagy-related genes (ATGs), such as *Atg3*, *Atg5*, *Atg7*, *Atg12*, and *Ulk1/2*, are born with normal morphology but die within 1 day after birth due to defective metabolism-autophagy crosstalk in response to nutrient stress[41]. *Ripk1*-deficient mouse neonates show a similar phenomenon[8–10], suggesting that RIPK1 is important in starvation resistance. In this work, we demonstrated that RIPK1 has an undefined function in regulating starvation resistance by modulating aspartate catabolism. This new regulatory role of RIPK1 is essential for energy homeostasis and autophagy in response to starvation. We found that the increased intracellular aspartate in $Ripk1^{-/-}$ cells enhanced the activity of the TCA cycle and increased the production of ATP in cells. Then, the disrupted energy metabolic balance caused defective autophagy induction, which contributed to the neonatal lethality of $Ripk1^{-/-}$ mice. These findings demonstrated that RIPK1 served as a metabolism regulator that is responsible for starvation resistance.

Starvation occurs not only in the neonatal period but also in ischemia/respiration, the tumor microenvironment, and other situations. Here, we also found that RIPK1 acts as a sensor of starvation and translates the starvation signal to transcription factor SP1 to modulate the transcription of aspartate catabolism genes. SP1 is a shared regulator of aspartate catabolism genes. Transcriptional analyses showed that RIPK1 deficiency inhibited the activity of the transcription factor SP1, resulting in down-regulated gene expressions in aspartate catabolism. Both SP1 overexpression and reduction of aspartate levels reversed the inhibitory effects of RIPK1 deficiency on starvation-induced autophagy. It is worth noting that, unlike the inhibition effect of *Ripk1* deficiency on starvation-induced autophagy, *Ripk1* deficiency promoted basal autophagy in normal conditions (Fig. 1j)[42]. We think that, although aspartate levels were increased in $Ripk1^{-/-}$ cells under both normal and starvation conditions, aspartate may have minimal effects on cell survival and autophagy under normal condition since amino acids are sufficient. However, under starvation condition, the increased aspartate levels inhibited the activation of autophagy and starvation resistance. Thus, RIPK1 seems to regulate autophagy by different mechanisms in normal growth condition and starvation condition.

Aspartate is an important intermediate metabolite required for anabolism and maintenance of redox homeostasis. Aspartate also serves as a building block of pyrimidine synthesis that is critical for cell proliferation[21,22]. Our results showed that aspartate was the specific metabolite that consistently responded to RIPK1 expression in both MEF/Jurkat cell lines and mouse tissues under starvation conditions (Fig. 2). Although we showed that aspartate metabolism is critical in RIPK1-regulated autophagy, by no means do we conclude that aspartate metabolism is the sole target of RIPK1. Other metabolites are consistently regulated by RIPK1 under different conditions (Fig. 2). It is likely that other downstream metabolic signaling pathways may also contribute to the biological functions of RIPK1.

RIPK1 is a multifunctional protein that regulates both cell death and survival. It has been reported that RIPK1 can regulate cell necrosis through energy metabolism[43]. Furthermore, Najafov et al. reported that RIPK1 regulates energy-sensing through the AMPK-mTOR pathway[44]. Here, we report that the RIPK1-aspartate catabolism-TCA cycle metabolic axis represents the underlying metabolic mechanism by which RIPK1 regulates the AMPK pathway and autophagy. Mechanistically, we found that RIPK1 regulates aspartate metabolism by inhibiting SP1 nuclear expression, probably by altered phosphorylation (T739) of SP1, in a kinase-independent manner. It has been well reported that some of RIPK1's functions, such as promotion of cell death, require its kinase activity[45], whereas others, such as the activation of MAPK and NF-κB[46] to mediate pro-survival signals, are kinase activity-independent. Here, our results suggest that the kinase activity of RIPK1 is dispensable for starvation resistance and autophagy. Although our data indicated that RIPK1 interacted with SP1 in cells, the mechanism by which RIPK1 regulates SP1 activity and phosphorylation in a kinase-independent manner requires more experimental evidences. Together, our results confirmed that RIPK1 serves as an important regulator to maintain energy homeostasis under nutrient stress and supports survival. Our study will facilitate future work to explore the functions of RIPK1 related to cellular metabolism. This also offers new possibilities for therapeutic applications by targeting RIPK1 to treat nutrient stress-related diseases[47].

## Methods

**Materials**. Cell line names and sources are described in Supplementary Table 3; recombinant DNA plasmids and their gene numbers are described in Supplementary Table 4; chemical names and catalog numbers are described in

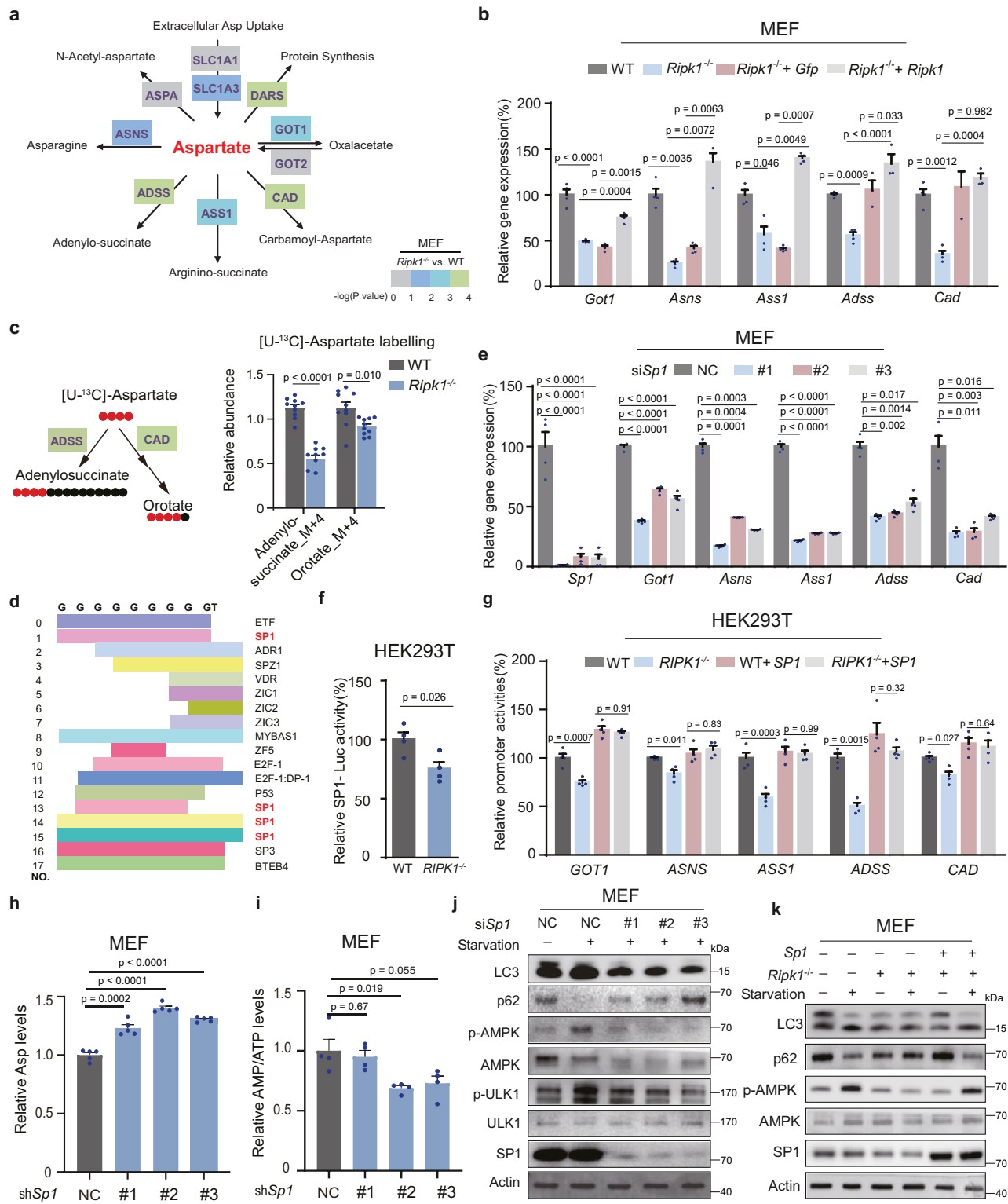

Supplementary Table 5; antibody names and catalog numbers are described in Supplementary Table 6.

**Mice.** Mice (C57BL/6) were group-housed in a barrier facility at room temperatures of 20–26 °C with 40–70% humidity and 12 h light/12 h dark cycles. All mice received a regular chow diet ad libitum. All animal care and experiments were conducted in accordance with the Institutional Animal Care and Use Committees of Interdisciplinary Research Center on Biology and Chemistry, Shanghai Institute of Organic Chemistry, Chinese Academy of Sciences.

**Electron microscopy.** Mice were perfused first with PBS and then 4% paraformaldehyde (PFA) and 1% glutaraldehyde for 4 h at 25 °C, and then postfixed in 1% OsO₄ followed by 2% uranyl acetate. After ethanol and propylene oxide dehydration, tissues were embedded in polybed 812 resin (Polysciences, 025950-1). Then ultrathin tissue sections (~70 nm) of brain and heart tissues were cut and post-stained with 2% uranyl acetate followed by 0.3% lead citrate. Tissue sections were analyzed on a TECNAI 10 transmission electron microscope (FEI) at an acceleration voltage of 80 kV. Autophagosomes typically have a double membrane. This structure is distinctly visible by TEM as two parallel membrane layers

**Fig. 6 RIPK1 deficiency suppresses aspartate catabolism by inactivating SP1. a** RNA-seq analysis of aspartate catabolic enzyme genes, including *Got1*, *Asns*, *Ass1*, *Adss*, *Cad*, *Dars* (Aspartateartyl-tRNA synthetase), *Slc1a1*, *Slc1a3*, *Aspa* in WT, and *Ripk1*[−/−] MEFs. The $-\log_{10}(P$ value) of genes (*Ripk1*[−/−] vs. WT) were indicated by box colors ($n = 4$ biologically independent samples per group). **b** Real-time PCR analyses of aspartate catabolic enzyme encoding genes *Got1*, Asns, *Ass1*, *Adss*, *Cad* in WT, *Ripk1*[−/−], *Ripk1*[−/−] + *Gfp*, and *Ripk1*[−/−] + *Ripk1* MEFs. Cells were cultured in EBSS for 4 h ($n = 4$ biologically independent samples per group). *P* values were determined by one-way ANOVA by Tukey's multiple comparisons test. **c** Relative abundances of adenylosuccinate and orotate in aspartate catabolism pathway ($n = 10$ biologically independent samples per group). Cells were cultured with 375 μM [U$^{13}$-C]-aspartate in EBSS for 7 h before collection. *P* values were determined by a two-tailed Student's *t*-test. **d** Motif scan for transcription factor prediction of the common GC-rich sequence in encoding genes of aspartate catabolic enzyme including *Got1*, *Cad*, *Ass1*, *Adss*, and *Asns*. **e** Real-time PCR analyses of *Sp1* and aspartate catabolic enzyme encoding genes responding to *Sp1* knockdown. MEFs were transfected with nontarget control (NC) or *Sp1* siRNAs (#1–3) ($n = 4$ biologically independent samples per group). Cells were cultured in EBSS for 4 h before collection. *P* values were determined by one-way ANOVA by Dunnett's multiple comparisons test. **f** Transcription activities of SP1 were determined by luciferase assay. WT and *RIPK1*[−/−] HEK293T cells were transfected with the *SP1*-luciferase (*Luc*) reporter plasmid for 24 h and cultured in EBSS for 4 h before harvest. The luciferase activities in WT cells were arbitrarily set to 100%, and the relative luciferase activities in the *RIPK1*[−/−] cells were calculated accordingly ($n = 4$ biologically independent samples per group). *P* values were determined by a two-tailed Student's *t*-test. **g** Luciferase analyses of relative promoter activities of aspartate catabolic enzyme encoding genes. WT and *RIPK1*[−/−] HEK293T cells were transfected with or without recombinant *SP1*-luciferase (*Luc*) reporter plasmids. Cell extracts were transfected for 24 h and starved in EBSS for 4 h before luciferase activity measurements ($n = 4$ biologically independent samples per group). *P* values were determined by one-way ANOVA by Tukey's multiple comparisons test. **h, i** Relative abundances of aspartate and intracellular AMP/ATP ratios in MEFs transfected with nontarget control (NC) or *Sp1* shRNAs(#1–3) measured using LC-MS. Cells were starved in EBSS for 4 h before collection (**h**, $n = 5$ biologically independent samples per group; **i**, $n = 4$ biologically independent samples per group). *P* values were determined by a two-tailed Student's *t*-test. **j** Western blot analyses of LC3, p62, p-AMPK, AMPK, p-ULK1, and ULK1 responding to *Sp1* knockdown. MEFs were transfected with nontarget control (NC) or *Sp1* siRNAs (#1–3). Cells were starved in EBSS for 4 h before collection. **k** Western blot analyses of LC3, p62, p-AMPK, and AMPK responding to *Sp1* overexpression in WT and *Ripk1*[−/−] MEFs. Cells were starved in EBSS for 4 h before collection. Bar graphs represent mean ± SEM.

(bilayers) separated by a relatively narrower or wider electron-translucent cleft. Autolysosomes typically have only one limiting membrane. Frequently, they contain electron-dense cytoplasmic material and/or organelles at various stages of degradation. Autolysosomes are typically smaller than autophagosomes, and their structures are typically stained dark in TEM samples.

**Immunohistochemistry**. Serial coronal brain or heart tissue sections (12 μm) were used for immunohistochemical staining with the following modifications. Briefly, coronal sections were blocked in Tris Buffered Saline (TBS) combined with 1% Triton X-100 and 10% donkey serum for 2 h. All primary antibodies were used and incubated for 2 days at 4 °C. Secondary antibodies were used and incubated for 4 h at room temperature. Fluorescently stained sections were counterstained with DAPI (200 ng/ml; Sigma-Aldrich) for 2–5 min, and then coverslipped with Gel/Mount (Biomeda).

**Cell imaging and statistical analysis**. Cells were fixed with 4% PFA (Sigma-Aldrich) and stained with 3 μg/mL DAPI (Beyotime). Image data quantification was measured using an ArrayScan HCS 4.0 reader with a 203 objective (Cellomics ArrayScan VTI) for DAPI-labeled nuclei and GFP-tagged intracellular proteins.

**Metabolites extraction**. Cell samples were extracted using a MeOH:ACN:H$_2$O (2:2:1, v/v) solvent mixture. A volume of 1 mL of cold solvent was added to each cell pellet, vortexed for 30 s, and incubated in liquid nitrogen for 1 min. The samples were then allowed to thaw at room temperature and sonicated for 10 min. This freeze-thaw cycle was repeated three times in total. To precipitate proteins, samples were incubated for 1 h at −20 °C, followed by 15 min of centrifugation at 17,000$g$ and 4 °C. The resulting supernatant was taken and evaporated to dryness in a vacuum concentrator. The dry extracts were then reconstituted in 100 μL of ACN:H$_2$O (1:1, v/v), sonicated for 10 min, and centrifuged for 15 min at 17,000$g$ and 4 °C to remove insoluble debris. The supernatant was then transferred to HPLC vials and kept at −80 °C until LC–MS analysis.

**Targeted metabolomics**. The metabolomics analyses of MEFs and Jurkat cells (WT vs. *Ripk1*[−/−]) under normal condition followed our previous publication[24]. Briefly, an HPLC system (1260 series, Agilent Technologies, USA) coupled to a triple quadrupole mass spectrometer (Agilent 6460, Agilent Technologies, USA) was used for LC–MS analyses. Phenomenex Luna aminopropyl column [particle size, 3 μm; 100 mm (length) × 2.1 mm (i.d.)] was used for separation. The metabolomics analyses of MEFs and Jurkat cells (WT, *Ripk1*[−/−], and *Ripk1*[−/−] + *Ripk1*) under starvation and mouse brain and liver tissues followed an updated method in the publication[48]. Briefly, a UHPLC system (1290 series, Agilent Technologies, USA) coupled to a triple quadrupole mass spectrometer (Agilent 6495, Agilent Technologies, USA) was used for LC–MS analyses. A BEH Amide column [particle size,1.7 μm; 100 mm (length) × 2.1 mm (i.d.)] was used for separation. A total of 200 metabolites were simultaneously monitored, and the positive/negative polarity switching was used. All metabolite identifications and quantification were further manually confirmed using MRM transitions and retention times from chemical

standards. The detailed instrument parameters were used the same as our previous reprot[48]. In Brief, Agilent MassHunter Workstation Data Acquisition (Version B.07.00), Agilent MassHunter Workstation Qualitative Analysis (Version B.07.00) were used in data collection.

**Isotope tracing analysis**. The MEFs were plated in 6-cm dishes at 2,000,000 cells/dish, and cultured in DMEM medium containing dFBS (10%) and penicillin/streptomycin (1%). The MEFs were first grown to 75–80% confluence in log-phase in the cell culture plates. Then, the culture medium solution was changed to a fresh medium solution with 375 μM [U-$^{13}$C]-aspartate or 4 mM [U-$^{13}$C]-glutamine. The labeling experiments were performed for 7 h for [U-$^{13}$C]-aspartate and 12 h for [U-$^{13}$C]-glutamine. Fast extraction of intracellular metabolites was performed as follows. In brief, the culture medium was quickly removed, and the cells were washed with the cold PBS twice. The cell dishes were placed on dry ice and the metabolite extraction solution (MeOH:ACN:H$_2$O = 2/2/1, v/v/v, 1 mL) was added to the dishes to quench the metabolism. The extraction solution was precooled at −80 °C for 1 h prior to the extraction. The plates were then incubated at −80 °C for at least 40 min. The cell contents were scraped and transferred to a 1.5-mL Eppendorf tube. The samples were vortexed for 1 min and centrifuged for 10 min at 17,000 x $g$ and 4 °C to precipitate the insoluble materials. The supernatant was taken to a new 1.5-mL Eppendorf tube and evaporated to dryness at 4 °C using a vacuum concentrator. The dried extracts were then reconstituted in 100 μL of ACN:H$_2$O (1:1, v/v), sonicated for 10 min, and centrifuged for 15 min at 17,000 x $g$ and 4 °C to remove insoluble debris. The supernatant was then transferred to HPLC vials and kept at −80 °C until LC–MS analysis. The LC–MS analysis was performed using a UHPLC system (1290 series; Agilent Technologies, USA) coupled to a quadrupole time-of-flight mass spectrometer (TripleTOF 6600, SCIEX, Canada).

**Measurements of AMP/ATP**. Intracellular AMP and ATP levels were measured using AMP-Glo™ Assay (Promega) and CellTiterGlo™ ATP assay (Promega) according to the manufacturer's instruction.

**Measurement of cellular OCR and ECAR**. Cellular OCR and ECAR values were measured using the Seahorse XFe96 Extracellular Flux Analyzer (Agilent Technologies, USA) following the manufacturer-supplied protocols. Briefly, MEFs were seeded in an XFe96-well microplate with $1.5 \times 10^4$ cells per well and incubated for 20 h in normal growth conditions then starved in EBSS for 4 h prior to XF assay. Oligomycin (14 mM; ATP synthase inhibitor), carbonyl cyanide-4-(trifluoromethoxy) phenylhydrazone (FCCP, 10 mM; cellular uncoupler), rotenone/antimycin A (4 mM each; complex 1/complex 3 inhibitor) were sequentially added to determine basal-, ATP-dependent-, maximal-, and mitochondria-independent oxygen consumptions, respectively. ECAR was measured in XF medium in basal conditions. Glucose (10 mM), oligomycin (5 μM), and 2-deoxyglucose (2-DG, 100 mM; glycolysis inhibitor) were sequentially added to measure glycolysis, glycolysis capacity, and glycolysis reverse, respectively. The Seahorse measurements were all performed with the following assay conditions: 3 min of mixing; 3 min of waiting; and 3 min of measurements. The reported OCR and ECAR values were

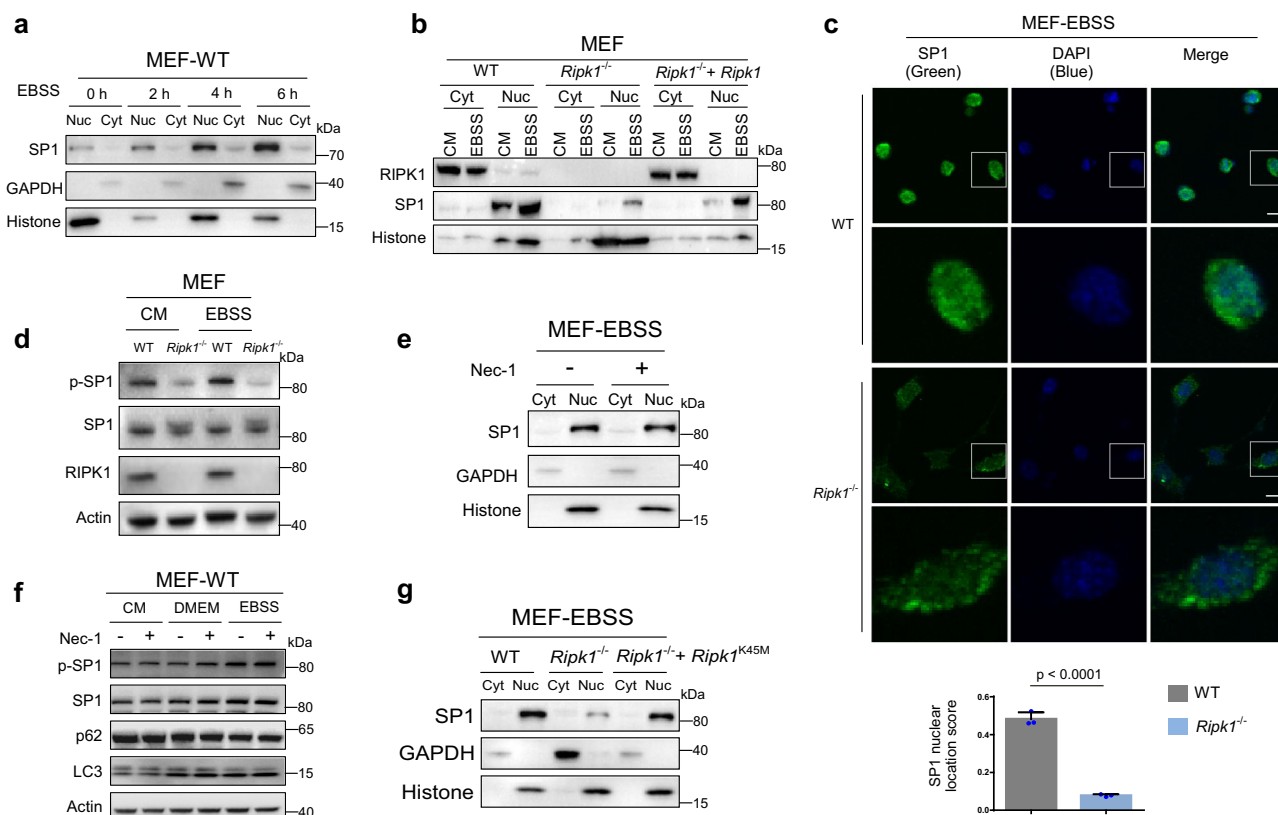

**Fig. 7 RIPK1 deficiency inactivates SP1 by inhibiting SP1 nuclear translocation. a** Nuclear (nuc) and cytosol (cyt) proteins of MEFs cultured in EBSS for 0, 2, 4, or 6 h were extracted and used to determine SP1 subcellular levels by western blot. **b** Cytosol (cyt) and nuclear (nuc) proteins of WT, *Ripk1*−/−, and *Ripk1*−/− + *Ripk1* MEFs were extracted and used to determine SP1 subcellular levels by western blot. The medium was changed to the fresh culture medium (CM) or EBSS for 4 h before harvest. **c** Representative immunofluorescence images depicting the levels and localizations of Sp1 (green) in WT, *Ripk1*−/− and *Ripk1*−/−+*Ripk1* MEFs. Cells were cultured in EBSS for 4 h before collection. The nuclei were stained with DAPI (blue). Scale bar represents 10 μm. Quantification of the nuclear staining of SP1 is presented in the bar graph. SP1 nuclear location score was computed as (the green and blue overlapping area)/(total green area + total blue area). The results were averaged from three independent immunofluorescence images. Bar graphs represent mean ± SD. **d** Western blot analyses of p-SP1 (T739) and SP1 levels in WT and *Ripk1*−/− MEFs. The medium was changed to the fresh culture medium (CM) or EBSS 4 h before harvest. **e** Cytosol (cyt) and nuclear (nuc) proteins of Nec-1 (100 μM, 24 h) treated or untreated MEFs were extracted and used to determine SP1 subcellular levels by western blot. The medium was changed to EBSS with or without Nec-1 4 h before harvest. **f** Western blot analyses of p-SP1 (T739), SP1, p62, and LC3 levels responding to Nec-1 treatment. The medium was changed to the fresh culture medium (CM), DMEM (serum-free), or EBSS for 4 h before harvest. **g** Cytosol (cyt) and nuclear (nuc) proteins of WT, *Ripk1*−/−, and *Ripk1*−/− + *Ripk1*^K45M MEFs were extracted and used to detect SP1 levels.

normalized to the cell numbers. The results were analyzed and exported from software Seahorse Wave (Version2.4.0.60).

**RNA-sequencing**. RNA was isolated from WT and *Ripk1*−/− MEFs. Residual DNA contamination was removed using a TURBO DNA-free kit according to the manufacturer's instruction (Thermo Fisher Scientific, USA). One microgram of total RNA was used for the preparation of the sequencing library. PolyA-tailed RNAs were selected by NEBNext Poly (A) mRNA Magnetic Isolation Module (NEB), followed by the library preparation using NEBNext Ultra RNA Library Prep Kit for Illumina according to the manufacturer's instruction. The libraries were checked by Bioanalyzer 2100 (Agilent Technologies, USA), then pooled together in equimolar amounts to a final concentration of 2 nM. Pooled denature libraries were sequenced on an Illumina MiSeq System.

**Real-time PCR**. RNA was isolated using Trizol Reagent (Thermo Fisher Scientific, USA) in accordance with the manufacturer's instructions. RNA was resuspended in DEPC-treated RNase-free water (Thermo Fisher Scientific, USA). Reverse transcription was catalyzed using a SuperScript III First-strand synthesis system (Thermo Fisher Scientific, USA). Real-time PCR analysis was performed using the QuantStudio 6 Flex real-time PCR system with SYBR selected master mix (Thermo Fisher Scientific, USA).

**Luciferase activity assay**. The reporter plasmids were transiently transfected into cells using the Lipofectamine method (Thermo Fisher Scientific, USA) according to

the manufacturer's protocol. Transfection efficiency was determined using the *Renilla* luciferase gene-containing pRL-CMV plasmid (Promega Corp.). HEK293T cells were transiently transfected with the reporter plasmid. After 24 h of the treatment, the transfected cells were washed twice with phosphate-buffered saline and lysed in passive lysis buffer (Promega Corp.) with gentle shaking at room temperature for 20 min. The cell lysate was centrifuged at 17,000xg for 2 min to pellet the cell debris. The supernatants were transferred to a fresh tube, and the dual-luciferase activity in the cell extracts was determined according to the manufacturer's protocol (Promega Corp.) The firefly luciferase activity was measured using a luminometer which was programmed to perform a 2 s pre-measurement delay, followed by a 10 s measurement period for each reporter assay. After measuring the firefly luciferase activity (Stop & Glo®, Promega Corp.), *Renilla* luciferase-measuring buffer was added, and the *Renilla* luciferase activity was measured. Each transfection was performed in duplication, and all measurements were repeated at least four times.

**Statistics and reproducibility**. All quantifications of fluorescence were represented as mean ± SEM from three replicates. Quantifications of immunoblots and immunostaining were performed using ImageJ and CellProfiler. All statistical analyses were performed using GraphPad Prism (v 6.0 and v 8.0) and Microsoft Excel 2010. All western blots from cell samples and mice tissues were repeated at least three times with similar results, with uncropped blots shown in the Source data. Differences were considered to be significant if *p* value <0.05.

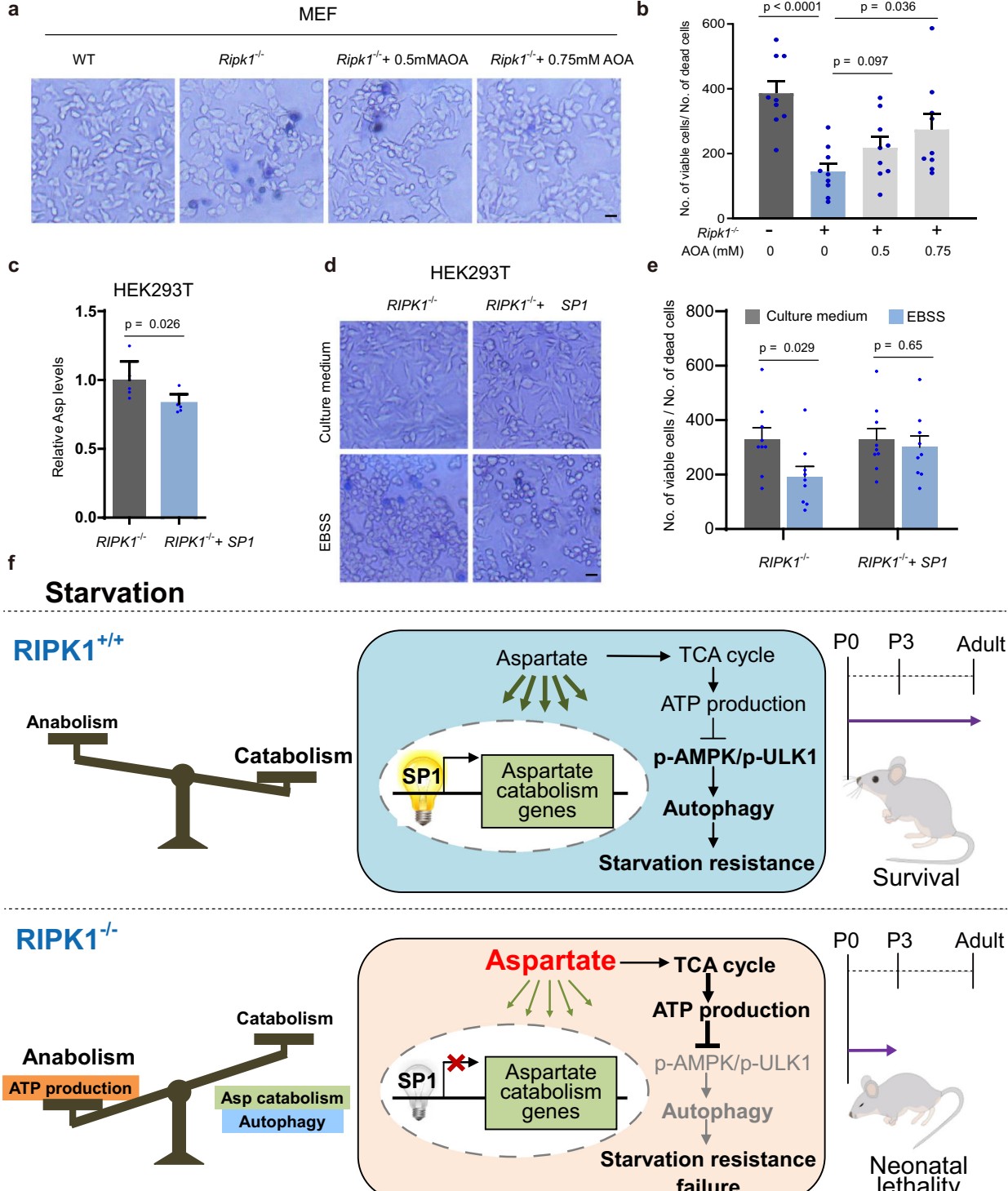

**Fig. 8 Reduced aspartate levels rescue the survival of *Ripk1*⁻/⁻ cells from starvation. a, b** Cell survival of *Ripk1*⁻/⁻ MEFs responding to AOA treatment was assessed by trypan blue exclusion. WT and *Ripk1*⁻/⁻ MEFs were cultured in EBSS for 6 h with 0.5 mM or 0.75 mM AOA treatment as indicated. Quantification of cell survival under different conditions was showed in (**b**). Data was expressed as the rate of viable cell number to dead cell number. Bars represent mean ± SEM (*n* = 9, three biologically independent samples per group, three images in each biological repetition). Scale bar represents 10 μm. *P* values were determined by one-way ANOVA by Tukey's multiple comparisons test. **c** Intracellular aspartate levels in *RIPK1*⁻/⁻ HEK293T cells transfected with pcDNA-3.1 or pcDNA-3.1-*SP1* plasmids were detected by LC–MS. Bars represent mean ± SEM (*n* = 6 biologically independent samples per group). *P* values were determined by a two-tailed Student's *t*-test. **d, e** Cell survival of *RIPK1*⁻/⁻ HEK293T cells responding to SP1 overexpression was assessed by trypan blue exclusion. Cells were transfected with pcDNA-3.1 or pcDNA-3.1-*SP1* plasmids. The medium was changed to culture medium or EBSS for 6 h before harvest. Quantification of cell survival under different conditions was showed in (**d**). Data was expressed as the rate of viable cell number to dead cell number. Bars represent mean ± SEM (*n* = 9; three biologically independent samples per group; three images in each biological repetition). Scale bar represents 10 μm. *P* values were determined by a two-tailed Student's *t*-test. **f** Graphical summary illustrating the role of RIPK1 in starvation resistance.

**Reporting Summary**. Further information on research design is available in the Nature Research Reporting Summary linked to this article.

## Data availability

The raw data files of the transcriptomics and metabolomics data generated in this study have been deposited and are available in the National Omics Data Encyclopedia under accession code OEP002409. The metabolite identification and quantification results generated in this study are provided in Supplementary Data 1–6. The identification and quantification of genes from RNA-sequencing generated in this study are provided in Supplementary Data 7. siRNA sequences are described in Supplementary Data 8. Primer sequences used in this study are described in Supplementary Data 9. Source data are provided with this paper.

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

## Acknowledgements

We thank Prof. Jiahuai Han (Xiamen University, China) for kindly providing the *RIPK1*$^{-/-}$ HEK293T cells, and Prof. Michelle Kelliher (UMass medical school) for kindly providing the *Ripk1*$^{-/-}$ mice. This work was financially supported by grants from the National Natural Science Foundation of China (31801167, 31971356, and 92057114), and Shanghai Municipal Science and Technology Major Project (2019SHZDZX02). X.M. is also supported by Shanghai Municipal Science and Technology Commission (17YF1424200). Z.-J.Z. is supported by the Excellent Young Scholar Fund from the National Natural Science Foundation of China (22022411).

## Author contributions

Z.-J.Z., X.M. and Y.G. conceived the idea and designed the project. X.M. designed and conducted major animal and biological experiments and analyzed the data. Z.X.

performed some biological experiments during revision. Y.G. and Z.X. performed the metabolomics analysis and other LC-MS analyses. Y.Z. and X.M. performed the RNA-seq analysis, and X.W. and Y.G. analyzed the RNA-seq data. N.L. contributed to the RNA-seq experiment. D.X. provided the GFP-LC3-H4 cell line. Y.L. provided the *Ripk1*$^{-/-}$ MEF, *Ripk1*$^{-/-}$ Jurkat cells, and *Ripk1* plasmids. Z.-J.Z. and X.M. wrote the manuscript. Z.-J.Z. supervised the project.

## Competing interests

The authors declare no competing interests.
