## [Peer Review File · Nature Communications]

RIPK1 regulates starvation resistance by modulating aspartate catabolismREVIEWER COMMENTS

Reviewer #1 (Remarks to the Author):

Starvation resistance is a profound mechanism mammalian cells perform to support their survival in nutrient diminished conditions. Such nutrient stressed instances that evoke the need for starvation resistance pathways include, but are not limited to, the tumor microenvironment, fasting, ischemia, and the neonatal period prior to nutrient provision. Within these starvation incidences, autophagy is a common process employed to catabolize macromolecules and/or organelles to supply amino acids and promote metabolic homeostasis. Previously, postnatal lethality was observed in neonates deficient of RIPK1, but the reason for this was not clearly understood. In the study conducted by Mei et.al., the authors conclude that RIPK1 promotes autophagy during starvation. Mechanistically, RIPK1 regulates the activity of transcription factor SP1 which facilitates the transcription of aspartate catabolizing enzymes. When aspartate is catabolized during starvation, AMP/ATP ratios increase, and AMPK alongside ULK1 activity promote autophagy. This is an interesting paper with the primary novelty coming from the discovery of the link between RIPK1 and aspartate metabolism. Some of the other findings (e.g. the role of AMPK, links to TCA flux and ATP levels) have been described previously in similar contexts. In addition to these concerns regarding novelty of the findings, there are several issues that would need to be addressed in a revised manuscript.

Major:

1. The premise that RIPK1 $-/-$ MEFs are disproportionately sensitive to starvation is not well supported by the data. The magnitude of the difference in the presence of EBSS (Fig. 1b) is not large and it is unclear how sustainable this difference is over a time course. Additional experimentation to prove this fundamental point is necessary in a revised manuscript.

2. Respectfully, I disagree with the authors' assertion that aspartate is the only metabolite that is elevated in RIPK1 $-/-$ cells and then rescued with RIPK1 complementation. The magnitude of the observed change is minimal and other metabolites (e.g. fumarate, glutamate) seem to respond in a similar manner (Fig 2c).

3. In some experiments, the authors use p62 abundance and the presence of LC3 II as surrogates for autophagic flux. However, there are some cases where the relationship between aspartate metabolism and autophagy seems only marginally supported. For example, in Figure 3e, siRNA of Got2 results in very small changes in aspartate levels yet p62 is completely lost and LC3II does not change significantly in 3f. How do the authors explain this result given the minimal change in aspartate levels caused by Got2 siRNA? Is it possible the Got2 phenotype is a result of an off target effect of the siRNA and thus the link to aspartate is not substantiated?

4. The expected relationship between p62 and LC3II is uneven in many of the westerns which diminishes confidence in the conclusion. As an example, compare Figure 4e and 4h. The expected relationship is clear cut in 4e but not so in 4h.

5. The authors provide Western blots demonstrating successful knockdowns and/or silencing of RIPK1 in MEFs and H4-GFP-LC3 in figure 1. Yet, Western blots confirming a successful RIPK1 knockdown are not provided for HEK293T-RIPK1 $-/-$ -GFP-LC3 cells in figure 3. Similarly, the confirmation of GOT2 knockdown in Figure 4 is not provided.

6. The regulatory ability RIPK1 has on SP1 was well demonstrated in terms of promoting aspartate catabolism. The authors show that RIPK1 does not affect the abundance of SP1, but rather its activity. Yet how RIPK1 influences SP1 activity is not clearly stated. More information on the mechanism here could add to the novelty of the study.

7. Throughout the manuscript, the authors included the use of a couple of different mouse tissue

samples to depict the presence/absence of autophagy induction. It would be helpful to better justify the decision to investigate autophagy in these tissues.

Minor:

1. In figure 2H, mouse in "mouse liver" is spelled incorrectly.
2. A thorough review is strongly recommended to correct grammar and spelling errors.

Reviewer #2 (Remarks to the Author):

In the manuscript by Mei et al., the authors report that RIPK1 regulates starvation resistance through modulating aspartate catabolism. They authors describe that RIPK1 deficiency increases aspartate levels. These increased aspartate levels then enhance TCA cycle and ATP production and ultimately block the AMPK-ULK1 axis of autophagy induction. Furthermore, the authors report that RIPK1 deficiency down-regulates enzymes involved in aspartate catabolism by inactivating the transcription factor SP1. Generally, I think this is an important observation for the research field addressing the communication between autophagy, cell death signaling and metabolism. However, I definitely think that this manuscript needs revision in order to be acceptable for publication in NATURE COMMUNICATIONS, since central mechanistic insights are missing.

Major point:

1) The authors did extensive work to characterize the RIPK1-dependent regulation of metabolism and starvation resistance. However, the central mechanistic detail is lacking, i.e. how RIPK1 deficiency inactivates SP1, or vice versa how RIPK1 activates/controls SP1. Is this a phosphorylation-dependent process? Is RIPK1 kinase activity required? How is inactivation achieved if RIPK1 is absent? SP1 mRNA and protein levels are not affected by RIPK1, so how is SP1 activity regulated by RIPK1? Is it related to nuclear localization? I think answers to these questions are central to this manuscript.

Minor points:

- 1) This manuscript needs extensive revision of the usage of the English language.
- 2) Page 4, first paragraph of results: "Under starvation, cells activate anabolism (...) and suppress catabolic processes (...)". I think it should be the other way around.
- 3) Page 6: the authors state that only the elevation of aspartate levels in Ripk1^{-/-} cells was rescued by Ripk1 complementation in MEFs and Jurkat cells. However, in both cell lines individual metabolites behaved similarly; perhaps the authors can at least mention this.
- 4) Page 8, middle paragraph: in the description of figure 4g, the authors mention Ripk1^{-/-} MEFs. However, in the figure and the legend wt MEFs are described. Similarly, in figure 4h wt MEFs are shown.
- 5) Page 9, last paragraph: the authors comment on Slc1a3; however, in figure 6a SLC1A1 is depicted.
- 6) Figure 1e/f: how were autophagosomes distinguished from autolysosomes in TEM?
- 7) Figure 2g/figure S2c: why is there a difference for N-carbamoyl-L-aspartate in these two figures?
- 8) Figure 3a: in the legend Ripk1^{-/-} MEFs are described, but the label in the figure is "wt"
- 9) Figures 3f and 4e: in these two figures, the p62 accumulation is weaker in starved Ripk1^{-/-} MEFs than in figure 1j; how can this be explained?
- 10) Figure 6c: which genes were exactly analyzed? Only the ones mentioned in the legend, or additional genes? This should be clarified.
- 11) Figures 6e and 6f: these experiments are only shown for RIPK1^{-/-} HEKs; however, I recommend repeating these assays in the previously used MEFs and Jurkat cells.
- 12) Figure S1d: please show also a magnification for Ripk1^{+/-} cells.
- 13) Tables S1 and S2: please exactly indicate the origin of the cells and the plasmids ("Gift from..."). Include References if applicable.

Reviewer #3 (Remarks to the Author):

The manuscript by Mei X. et al focuses on the role of Ripk1 as regulator of starvation via modulation of cellular metabolism. Using multiple models (in vitro and in vivo) they find that cells knock-out for Ripk1 accumulate aspartate and fail to induce autophagy upon starvation. Mechanistically (i) autophagy is inhibited by a decrease in AMP/ATP ratio that leads to inhibition of AMPK and (ii) aspartate is accumulated because of inhibition of its catabolic pathways regulated by the transcription factor Sp1, which is found less active in Ripk1 ^{-/-} cells. Ultimately, reduction of the levels of aspartate in Ripk1 ^{-/-} cells helps the cells during starvation.

This is a timely and novel manuscript on a topic of clear interest. Experiments are properly performed and important controls are in place, such as the reconstitution of Ripk^{-/-} with Ripk1.

Main points of revision:

1. The data presented do support the statement of the authors that the level of aspartate in Ripk1 ^{-/-} cells inhibits autophagy upon starvation, however the data also show that in normal growth conditions aspartate level is higher and autophagy is induced in Ripk1 ^{-/-} as also pointed out by the authors (lines 88-89) and Fig 1 i-k. These two observations are contradictory and would need some explanation. Is autophagy induced by different mechanisms in normal growth conditions vs starvation? Along the same line it would be important to show/measure the level of aspartate upon starvation. Currently the data are shown only in normal growth conditions (Fig. 2c).

2. In Fig. 3 e-l the authors show that reducing the concentration of aspartate via Got2 siRNA or using the AOA inhibitor in Ripk1 ^{-/-} MEFs allows induction of autophagy upon starvation. This is an important set of data supporting the hypothesis, but it requires some further explanation. Indeed Got2 siRNA has a significant, but very small effect on the level of aspartate (Fig. 3e) and a very clear effect in inducing autophagy, see for example p62 WB in Fig. 3f; on the contrary, the use of AOA has a massive effect on aspartate (Fig 3i) but a very similar effect on autophagy induction (if not smaller, see for example WB in Fig. 3J). If the level of aspartate inhibits autophagy induction the effect should be proportional to its concentration?

3. The proposed mechanism for the increased level of aspartate in Ripk1 ^{-/-} is inhibition of its catabolism. A number of enzymes involved in aspartate catabolism are shown to be downregulated: GOT1, ASNS, ASS1 and CAD (Fig. 6). These data do support the hypothesis that aspartate catabolism is inhibited, but further evidences are required to fully prove this. For example, what are the levels of the metabolites produced by these reactions?

Also, in Fig 5l label aspartate is given to the cells (wt and Ripk1 ^{-/-}), how is it metabolized? Does it accumulate in cells as it should happen if its catabolism is inhibited?

Minor points:

1. The authors should indicate which statistical test have been utilised to calculate significance;
2. There are at least two published manuscripts on the role of RipK1 as regulator of metabolism: 10.1126/science.1172308 and 10.1016/j.molcel.2020.11.008. These two papers should be discussed, in particular the second one which is related to the topic covered in this manuscript.
3. Throughout the manuscript experiments are performed alternating normal growth conditions and starvation, not always with a clear rationale. It would be great to make the rationale clearer and also to indicate clearly in the figure in which conditions the experiment has been performed to help the reader.

Response to the reviewers:

The authors would like to thank the reviewers for the helpful comments. We believed that these comments have strengthened the manuscript considerably.

Reviewer #1:

General comment: “*Starvation resistance is a profound mechanism mammalian cells perform to support their survival in nutrient diminished conditions. Such nutrient stressed instances that evoke the need for starvation resistance pathways include, but are not limited to, the tumor microenvironment, fasting, ischemia, and the neonatal period prior to nutrient provision. Within these starvation incidences, autophagy is a common process employed to catabolize macromolecules and/or organelles to supply amino acids and promote metabolic homeostasis. Previously, postnatal lethality was observed in neonates deficient of RIPK1, but the reason for this was not clearly understood. In the study conducted by Mei et.al., the authors conclude that RIPK1 promotes autophagy during starvation. Mechanistically, RIPK1 regulates the activity of transcription factor SP1 which facilitates the transcription of aspartate catabolizing enzymes. When aspartate is catabolized during starvation, AMP/ATP ratios increase, and AMPK alongside ULK1 activity promote autophagy. This is an interesting paper with the primary novelty coming from the discovery of the link between RIPK1 and aspartate metabolism. Some of the other findings (e.g. the role of AMPK, links to TCA flux and ATP levels) have been described previously in similar contexts. In addition to these concerns regarding novelty of the findings, there are several issues that would need to be addressed in a revised manuscript.*”

Ans: We appreciate the positive comments to our study from the reviewer. Following the reviewer’s suggestion, we have made the required changes in the revised manuscript.

Comment #1: “*The premise that RIPK1^{-/-} MEFs are disproportionately sensitive to starvation is not well supported by the data. The magnitude of the difference in the presence of EBSS (Fig. 1b) is not large and it is unclear how sustainable this difference is over a time course. Additional experimentation to prove this fundamental point is necessary in a revised manuscript.*”

Ans: We appreciate the comment from the reviewer. The original data provided in Fig.1b was performed under EBSS starvation for 6 h. In the revised manuscript, we have followed the suggestion from the reviewer, and performed the starvation experiment over a time course (6 h, 12 h, and 24 h). The new data is provided in **Supplementary Fig. 1a** and shown below. Indeed, with the increased starvation time, survival differences between WT and *Ripk1^{-/-}* cells were enlarged.

Copy of Supplementary Fig. 1a.

(a) Assessments of cell survival by trypan blue exclusion. WT and *Ripk1*^{-/-} MEF cells were cultured in culture medium, glucose free medium, EBSS, or EBSS with CHX, respectively, for different times as indicated. Data were expressed as the ratios of viable cell numbers to total cell numbers (n = 3 biologically independent samples per group).

Comment #2: “Respectfully, I disagree with the authors’ assertion that aspartate is the only metabolite that is elevated in *RIPK1*^{-/-} cells and then rescued with *RIPK1* complementation. The magnitude of the observed change is minimal and other metabolites (e.g. fumarate, glutamate) seem to respond in a similar manner (Fig 2c).”

Ans: We thank the reviewer’s comment. We first profiled the metabolic changes responding to *Ripk1* deficiency in MEF cells under normal condition (Fig. 2a-c), and found that the alanine, aspartate and glutamate metabolism pathway was one of the enriched pathways. Indeed, as the reviewer pointed, other metabolites, such as fumarate and glutamate in this pathway, were also changed in response to *Ripk1* deficiency.

As suggested by Reviewer #3, the metabolic changes under starvation in response to *Ripk1* deficiency are more relevant to our study. Therefore, in revised manuscript, we further profiled the metabolic changes in starvation conditions in both MEF and Jurkat cells (Fig. 2d, e). In addition, metabolic changes in mouse neonate tissues (liver and brain) were also provided (Fig. 2f, g). Under starvation conditions, in all of the four data sets, we confirmed that aspartate was the only metabolite that is consistently responding to *RIPK1* expression (Fig. 2h). The new data were provided below.

Although our results showed that aspartate was the specific metabolite that consistently responded to *RIPK1* expression in both MEF/Jurkat cell lines and mouse tissues under starvation conditions (Fig. 2), by no means do we conclude that aspartate metabolism is the sole target of *RIPK1*. There were other metabolites that were consistently regulated by *RIPK1* under different conditions. It is likely that other downstream metabolic signaling pathways may also contribute to the biological functions of *RIPK1*. We have added the related discussion in the revised manuscript.

Finally, we also examined the changes of fumarate and glutamate under starvation conditions. Apparently, although fumarate and glutamate were increased in response to *Ripk1* deficiency in normal condition, they had no consistent changes under different starvation conditions (Fig. R1, see below). Therefore, in our study, fumarate and glutamate were not selected for mechanistic study.

Copy of Figure 2d-h in the revised manuscript.

(d, e) Metabolites were significantly increased/decreased in *Ripk1*^{-/-} group compared with WT group ($p < 0.05$), and significantly rescued ($p < 0.05$) in *Ripk1*^{-/-} + *Ripk1* group compared with *Ripk1*^{-/-} group: MEF cells (d); Jurkat cells (e). All cells were cultured in EBSS for 4 h. Bar graphs represent mean \pm SEM ($n = 7-10$ biologically independent samples in each group). P values were determined by two-tailed Student's t-test with FDR correction.

(f, g) Metabolites were significantly ($p < 0.05$) increased/decreased in *Ripk1*^{-/-} compared with WT, and significantly rescued ($p < 0.05$) in *Ripk1*^{+/-} compared with *Ripk1*^{-/-} in mouse liver (f) and brain (g) under starvation. Bar graphs represent mean \pm SEM ($n = 5-6$ biologically independent samples in each group). P values were determined by two-tailed Student's t-test with FDR correction.

(h) Venn diagram of the overlapped RIPK1-dependent metabolites in d (MEF cells), e (Jurkat cells), f (Mouse liver) and g (Mouse brain) under starvation conditions.

Figure R1. The abundance changes of fumarate (a) and glutamine (b) in MEF, Jurkat, mouse brain and mouse liver under starvation conditions.

Comment #3. “In some experiments, the authors use p62 abundance and the presence of LC3 II as surrogates for autophagic flux. However, there are some cases where the relationship between aspartate metabolism and autophagy seems only marginally supported. For example, in Figure 3e, siRNA of *Got2* results in very small changes in aspartate levels yet p62 is completely lost and LC3II does not change significantly in 3f. How do the authors explain this result given the minimal change in aspartate levels caused by *Got2* siRNA? Is it possible the *Got2* phenotype is a result of an off target effect of the siRNA and thus the link to aspartate is not substantiated?”

Ans: We thank the reviewer’s careful examination. The transfection efficiency of MEF cells is relatively low and not consistent in different batches. And in the original data, the LC-MS data and WB data were not collected from the same batch of cell samples. Therefore, the original data were not quantitatively supported as observed by the reviewer.

In the revised manuscript, we prepared stable GOT2 knockdown cell lines and repeated the LC-MS and WB analyses using the cell samples from the same batch. The new data were provided in **Fig. 3e** and **3f** in the revised manuscript (see below). As you will see, all of three GOT2 shRNAs successfully reduced the expressions of GOT2 in protein levels. Among them, #1 shRNA induced the most significant decrease of GOT2 level (**Fig. 3f**). Accordingly, we consistently observed the most significant decrease of aspartate level and increase of autophagy in *Ripk1*^{-/-} cells with #1 shRNA (**Fig. 3e**).

Copy of Figure 3e and f.

(e) Intracellular aspartate levels in *Ripk1*^{-/-} MEF cells were measured using LC-MS. *Ripk1*^{-/-} MEF cells were transfected with non-target control (NC) or *Got2* shRNAs (#1-3), and cultured in EBSS for 4 h (n = 5 or 6 biologically independent samples per group).

(f) Western blot analyses of LC3, p62 and GOT2 in *Ripk1*^{-/-} MEF cells.

Comment #4. “The expected relationship between p62 and LC3II is uneven in many of the westerns which diminishes confidence in the conclusions. As an example, compare Figure 4e and 4h. The expected relationship is clear cut in 4e but not so in 4h.”

Ans: Thanks for the reviewer’s comment. We think the issue in **Fig.4h** might be due to the defective WB analysis, which generated the underestimated levels of p62. We checked our records and found the duplicated data record from the same sample batch. The p62 levels of these samples were also examined in another WB analysis. Although the p62 signals were not very clear, we can see that the levels of p62 and LC3-II matched well with each other in different conditions. In the revised manuscript, Fig. 4h has been updated (see below).

Copy of Figure 4h

(h) Western blot analyses of LC3, p62, p-AMPK, AMPK, p-ULK1 and ULK1 levels in WT MEF cells treated with AICAR (0.5 mM, 4 h) and aspartate (0.15 mM; 4 h) in EBSS as indicated.

Comment #5. “The authors provide Western blots demonstrating successful knockdowns and/or silencing of RIPK1 in MEFs and H4-GTP-LC3 in figure 1. Yet, Western blots confirming a successful RIPK1 knockdown are not provided for HEK293T-RIPK1^{-/-}-GFP-LC3 cells in figure 3. Similarly, the confirmation of GOT2 knockdown in Figure 4 is not provided.”

Ans: We appreciate the comment from the reviewer. In the revised manuscript, we have provided the RIPK1 knockdown data in the revised Fig. 3m, and GOT2 knockdown data in the revised Fig. 4j.

Fig. 3m

Fig. 4j

Copy of Figure 3m and 4j

(3m) Western blot analyses of RIPK1 and Actin showed the knockdown efficiency of RIPK1 in (k) and (l).

(4j) Western blot analyses of p-AMPK, AMPK, p-ULK1 and ULK1 levels in *Ripk1*^{-/-}MEF cells in response to *Got2* knockdown. *Got2* knockdown efficiency was detected using GOT2 antibodies. Cells were transfected with NC or *Got2* siRNAs (#1-3), cultured in normal medium or EBSS for 4 h.

Comment #6. “The regulatory ability RIPK1 has on SP1 was well demonstrated in terms of promoting aspartate catabolism. The authors show that RIPK1 does not affect the abundance of SP1, but rather its activity. Yet how RIPK1 influences SP1 activity is not clearly stated. More information on the mechanism here could add to the novelty of the study.”

Ans: Thanks for the reviewer’s suggestion. In the revised manuscript, we further investigated how RIP1 influences SP1 activity. Transcriptional activity of SP1 could be affected by many aspects, such as binding affinity to promoters, nuclear translocation, protein stability, interaction with other protein factors, and others. As suggested by other reviewers, we specifically investigated whether RIPK1 influences the nuclear translocation of SP1. The results demonstrated that, in WT MEF cells, SP1 was largely enriched in nuclear, while SP1 nuclear translocation was significantly decreased in *Ripk1*^{-/-} MEF cells under starvation (Fig. 7a-c). SP1 nuclear translocation is regulated by its phosphorylation. We also found that the SP1 phosphorylation (T739) was significantly decreased in *Ripk1*^{-/-} MEF cells (Fig. 7d). We also confirmed that RIPK1 kinase activity is not required for SP1 nuclear translocation and phosphorylation (Fig. 7e and 7f). Transfection of the RIPK1 kinase-dead mutant (K45M) successfully rescued SP1 nuclear localization in *Ripk1*^{-/-} MEFs (Fig. 7g). Finally, we demonstrated that RIPK1 and SP1 had colocalizations in cells and confirmed interactions by co-IP experiments (Supplementary Figure S7). All of the related data are provided in Figure 7 and Supplementary Figure S7.

Copy of Figure 7. RIPK1 deficiency inactivates SP1 by inhibiting SP1 nuclear translocation

(a) Nuclear (nuc) and Cytosol (cyt) proteins of MEF cells cultured in EBSS for 0 h, 2 h, 4 h and 6 h were extracted and used to determine SP1 subcellular levels by western blot.

(b) Cytosol (cyt) and nuclear (nuc) proteins of WT, *Ripk1*^{-/-} and *Ripk1*^{-/-} + *Ripk1* MEF cells were extracted and used to determine SP1 subcellular levels by western blot. The medium was changed to fresh culture medium (CM) or EBSS for 4 h before harvest.

(c) Representative immunofluorescence images depicting the levels and localization of Sp1 (green) in WT, *Ripk1*^{-/-}, and *Ripk1*^{-/-} + *Ripk1* MEF cells. Cells were cultured in EBSS for 4 h before collection. The nuclei were stained with DAPI (blue). Scale bar = 10 μm.

(d) Western blot analyses of p-SP1 (T739) and SP1 levels in WT and *Ripk1*^{-/-} MEF cells. The medium was changed to fresh culture medium (CM) or EBSS 4 h before harvest.

(e) Cytosol (cyt) and nuclear (nuc) proteins of Nec-1 (100 μM, 24 h) treated or untreated MEF cells were extracted and used to determine SP1 subcellular levels by western blot. The medium was changed to EBSS with or without Nec-1 4 h before harvest.

(f) Western blot analyses of p-SP1 (T739), SP1, p62 and LC3 levels responding to Nec-1 treatment. The medium was changed to fresh culture medium (CM), DMEM (serum free), or EBSS for 4 h before harvest.

(g) Cytosol (cyt) and nuclear (nuc) proteins of WT, *Ripk1*^{-/-} and *Ripk1*^{-/-} + *Ripk1*^{K45M} MEF cells were extracted and used to detect SP1 levels.

Copy of Supplementary Figure 7. RIPK1 co-located and interacted with SP1

(a) Representative immunofluorescence images depicting the localizations of SP1 (green) in Nec-1 treated or untreated MEFs. Cells were cultured in culture medium (CM) or EBSS for 6 h before collection. The nuclei were stained with DAPI (blue). Scale bar = 10 μ m. White boxes indicate the magnified regions.

(b) Cytosol (cyt) and nuclear (nuc) proteins of Nec-1 (100 μ M, 24 h) treated or untreated MEF cells were extracted and used to determine SP1 subcellular levels by western blot. Cells were cultured in culture medium (CM).

(c, d) Representative immunofluorescence images depicting the co-localization of RIPK1 (green) and SP1 (red, **c**) or p-SP1T739 (red, **d**) in MEFs. Cells were cultured in culture medium (CM) or EBSS for 6 h before collection. The nuclei were stained with DAPI (blue). Scale bar = 5 μ m.

(e) Co-immunoprecipitation of SP1 and RIPK1. Flag-SP1, HA-RIPK1 and vector plasmids were transfected in HEK293T cells as indicated. Cells were starved in EBSS for 4 h. Immunoprecipitation was performed using flag-beads.

Comment #7. *“Throughout the manuscript, the authors included the use of a couple of different mouse tissue samples to depict the presence/absence of autophagy induction. It would be helpful to better justify the decision to investigate autophagy in these tissues.”*

Ans: Thanks for the reviewer’s comment. We chose these tissues by referring to the publication in *Nature* (*Nature* 432, 1032–1036 (2004), <https://doi.org/10.1038/nature03029>). This paper reported that autophagy is important for postnatal survival during neonatal starvation, and demonstrated that the formation of autophagosomes was extensively induced in various tissues including heart, brain and liver tissues after birth. In particular, the heart displayed massive autophagy. Therefore, in our work, we also chose heart, brain, and liver tissues to investigate the autophagy. We have added the related information in the revised manuscript.

Comment #8. *“In figure 2H, mouse in “mouse liver” is spelled incorrectly.”*

Ans: We thank the reviewer’s comment, we have corrected the spelling in the revised manuscript.

Comment #9. *“A thorough review is strongly recommended to correct grammar and spelling errors.”*

Ans: Thanks for your suggestion. We have used the Nature Research Editing Service to improve the language quality of our manuscript.

Reviewer #2:

General comment: “In the manuscript by Mei et al., the authors report that RIPK1 regulates starvation resistance through modulating aspartate catabolism. They authors describe that RIPK1 deficiency increases aspartate levels. These increased aspartate levels then enhance TCA cycle and ATP production and ultimately block the AMPK-ULK1 axis of autophagy induction. Furthermore, the authors report that RIPK1 deficiency down-regulates enzymes involved in aspartate catabolism by inactivating the transcription factor SP1. Generally, I think this is an important observation for the research field addressing the communication between autophagy, cell death signaling and metabolism. However, I definitely think that this manuscript needs revision in order to be acceptable for publication in NATURE COMMUNICATIONS, since central mechanistic insights are missing.”

Ans: We appreciate the reviewer’s careful reading and positive assessments of our manuscript.

Comment #1. “The authors did extensive work to characterize the RIPK1-dependent regulation of metabolism and starvation resistance. However, the central mechanistic detail is lacking, i.e. how RIPK1 deficiency inactivates SP1, or vice versa how RIPK1 activates/controls SP1. Is this a phosphorylation-dependent process? Is RIPK1 kinase activity required? How is inactivation achieved if RIPK1 is absent? SP1 mRNA and protein levels are not affected by RIPK1, so how is SP1 activity regulated by RIPK1? Is it related to nuclear localization? I think answers to these questions are central to this manuscript.”

Ans: Thanks for the reviewer’s suggestion. In the revised manuscript, we further investigated how RIPK1 influences SP1 activity (**Figure 7 and Supplementary Figure S7 in Page R6-8**). As suggested by the reviewers, we specifically investigated whether RIPK1 influences the nuclear translocation of SP1. The results demonstrated that, in WT MEF cells, SP1 was largely enriched in nuclear, while SP1 nuclear translocation was significantly decreased in *Ripk1*^{-/-} MEF cells under starvation conditions (**Fig. 7a-c**). SP1 nuclear translocation is regulated by its phosphorylation. We found that the SP1 phosphorylation (T739) was significantly decreased in *Ripk1*^{-/-} MEF cells (**Fig. 7d**). We also confirmed that RIPK1 kinase activity is not required for SP1 nuclear translocation. The treatment of MEF cells with Nec-1 had no effect on SP1 nuclear location, phosphorylation and activation of autophagy (**Fig. 7e and 7f**), while the kinase dead *Ripk1*^{K45M} was sufficient to rescue SP1 nuclear translocation (**Fig. 7g**). Our data also indicated that RIPK1 and SP1 had colocalizations and interactions. All related data are provided in **Figure 7 and Supplementary Figure S7**.

Comment # 2: “This manuscript needs extensive revision of the usage of the English language.”

Ans: Thanks for the suggestion. We have used the Nature Research Editing Service to improve the language quality of our manuscript.

Comment #3: “Page 4, first paragraph of results: “Under starvation, cells activate anabolism (...) and suppress catabolic processes (...).” I think it should be the other way around.”

Ans: Sorry for the mistake. We have revised this sentence.

Comment #4: “Page 6: the authors state that only the elevation of aspartate levels in *Ripk1*^{-/-} cells was rescued by *Ripk1* complementation in MEFs and Jurkat cells. However, in both cell lines individual metabolites behaved similarly; perhaps the authors can at least mention this.”

Ans: We thank the reviewer’s comment. In the initial metabolomic profiling, we measured the metabolic changes in response to *Ripk1* deficiency in MEF cells under normal condition, and found that the alanine, aspartate and glutamate metabolism pathway was one of the enriched pathways (**Fig. 2a-c**). In addition to aspartate, other metabolites, such as fumarate and glutamate in this pathway, were also changed in response to *Ripk1* deficiency in MEF cells.

However, the metabolic changes under starvation responding to *Ripk1* deficiency are more relevant to our study. Therefore, in revised manuscript, we further profiled the metabolic changes of both MEF and Jurkat cells in starvation conditions (**Fig. 2d, e**). We provided all significantly changed metabolites in *Ripk1*^{-/-} cells, and those were rescued by *Ripk1* complementation (**Fig. 2d, e**). The similar analyses were also performed for metabolomics analysis of mouse neonate tissues (liver and brain, **Fig. 2f, g**) under starvation. Finally, in all of the four data sets, we confirmed that aspartate was the only metabolite that is consistently responding to RIPK1 expression (**Fig. 2h**). All related data are provided in **Figure 2** and Supplementary **Figure S2** in the revised manuscript.

Comment #5 “Page 8, middle paragraph: in the description of figure 4g, the authors mention *Ripk1*^{-/-} MEFs. However, in the figure and the legend wt MEFs are described. Similarly, in figure 4h wt MEFs are shown.”

Ans: Sorry for our mistake. We have corrected the result description in the revise manuscript. The WT MEFs should be correct.

Comment #6 “Page 9, last paragraph: the authors comment on *Slc1a3*; however, in figure 6a *SLC1A1* is depicted.”

Ans: Sorry for our mistake. We checked our RNA-seq data and found that expression of *Slc1a1* did not have the significant change, while *Slc1a3* significantly decreased in *Ripk1*^{-/-} MEFs (See revised **Figure 6a** and **Supplementary Figure 6a**).

Comment #7: “Figure 1e/f: how were autophagosomes distinguished from autolysosomes in TEM?”

Ans: We thank the reviewer’s comment. Double-membrane vesicles were counted as autophagosomes and electron dense vesicular structures were counted as autolysosomes.

Comment #8: “Figure 2g/figure S2c: why is there a difference for N-carbamoyl-L-aspartate in these two figures?”

Ans: We appreciate the careful reading of the reviewer. After checking our original data, we found

that the original Figure 2g and Figure S2c were from two different data sets of Jurkat cells. And the metabolic profiling was performed under normal condition. When we were addressing the related comments from Reviewers 1 and 3, we realized that the metabolic changes under starvation in response to *Ripk1* deficiency are more relevant to our study. Therefore, in revised manuscript, we further profiled the metabolic changes of both MEF and Jurkat cells in starvation conditions (**Fig. 2d and 2e**). After revision, the original Fig. 2g has been updated to **Fig. 2e** (RIPK1-dependent metabolites in Jurkat cells under starvation condition).

In addition, to ensure the data reproductivity, we have provided the source data of **Figure 2** and **Supplementary Figure S2** with the submission. The metabolomics results in this study were also provided in Supplementary Data 1-6. The original metabolomics data files were deposited into the public database, and freely downloaded with the URL (<https://www.biosino.org/node/project/detail/OEP002409>).

Comment #9: *“Figure 3a: in the legend Ripk1^{-/-} MEFs are described, but the label in the figure is “wt”.”*

Ans: We appreciate the careful reading of the reviewer. We have corrected *Ripk1^{-/-}* MEFs to WT MEFs in the figure legend.

Comment #10: *“Figures 3f and 4e: in these two figures, the p62 accumulation is weaker in starved Ripk1^{-/-} MEFs than in figure 1j; how can this be explained?”*

Ans: We thank the reviewer's comment. The EBSS starvation time in Fig 3f and Fig. 4e was 4 h, while the EBSS starvation time in Fig. 1j was 6 h. The reason that p62 accumulation was more significant in Fig. 1j may be due to the stronger autophagy flux induced by the longer starvation.

Comment #11: *“Figure 6c: which genes were exactly analyzed? Only the ones mentioned in the legend, or additional genes? This should be clarified.”*

Ans: We thank the reviewer's comment. In this analysis, genes for aspartate metabolic enzymes were analyzed. We have provided the full list of genes in **Supplementary Table 1** in the revised manuscript.

Comment #12: *“Figures 6e and 6f: these experiments are only shown for RIPK1^{-/-} HEKs; however, I recommend repeating these assays in the previously used MEFs and Jurkat cells.”*

Ans: We thank the reviewer's comment. The luciferase assay experiment needs to transfect multiple plasmids. MEF cells are difficult for transfection. Therefore, we used WT and *Ripk1^{-/-}* HEK293T for the luciferase assay experiment. In the revised manuscript, although we did not repeat the luciferase assay experiment on MEF cell, we have confirmed that the regulatory role of RIPK1 on SP1 nuclear translocation and phosphorylation in MEF cells (**Figure 7 and Supplementary Figure S7**). We wish these data from two cell lines are enough to validate our conclusions.

Comment #13: “Figure S1d: please show also a magnification for *Ripk1*^{+/-} cells.”

Ans: Thanks for your suggestion, we have provided a magnification for *Ripk1*^{+/-} group in the revised **Supplementary Figure 1e** in the revised manuscript.

Copy of Supplementary Figure 1e

(e) Representative immunofluorescence staining images of brain tissues for LC3 dot intensities (green dots) and DAPI (blue) from WT, *Ripk1*^{-/-} and *Ripk1*^{+/-} mouse neonates at 0 h or 6 h after birth without milk feeding. Scale bar represents 200 μm. Grey boxes indicate the magnified regions. Scale bar in the magnified images represents 50 μm.

Comment #14: “Tables S1 and S2: please exactly indicate the origin of the cells and the plasmids (“Gift from...”). Include References if applicable.”

Ans: Thanks for your suggestion, we have revised the original tables as **Supplementary Tables S3 and S4** in the revised manuscript.

Reviewer #3:

General comment: “The manuscript by Mei X. et al focuses on the role of *Ripk1* as regulator of starvation via modulation of cellular metabolism. Using multiple models (*in vitro* and *in vivo*) they find that cells knock-out for *Ripk1* accumulate aspartate and fail to induce autophagy upon starvation. Mechanistically (i) autophagy is inhibited by a decrease in AMP/ATP ratio that leads to inhibition of AMPK and (ii) aspartate is accumulated because of inhibition of its catabolic pathways regulated by the transcription factor *Sp1*, which is found less active in *Ripk1*^{-/-} cells. Ultimately, reduction of the levels of aspartate in *Ripk1*^{-/-} cells helps the cells during starvation. This is a timely and novel manuscript on a topic of clear interest. Experiments are properly performed and important controls are in place, such as the reconstitution of *Ripk1*^{-/-} with *Ripk1*.”

Ans: We appreciate the reviewer’s positive assessments to our study.

Comment #1: “The data presented do support the statement of the authors that the level of aspartate in *Ripk1*^{-/-} cells inhibits autophagy upon starvation, however the data also show that in normal growth conditions aspartate level is higher and autophagy is induced in *Ripk1*^{-/-} as also pointed out by the authors (lines 88-89) and Fig 1 i-k. These two observations are contradictory and would need some explanation. Is autophagy induced by different mechanisms in normal growth conditions vs starvation? Along the same line it would be important to show/measure the level of aspartate upon starvation. Currently the data are shown only in normal growth conditions (Fig. 2c).”

Ans: We thank the reviewer’s great comment. It has been reported that *RIPK1* knockdown promotes basal autophagy (EMBO Reports, 2015, <https://doi.org/10.15252/embr.201439496>). In EMBO study, the authors reported that “*RIPK1* represses basal autophagy in part due to its ability to regulate the *TFEB* transcription factor, which controls the expression of autophagy-related and lysosomal genes”. Consistently, our data also showed that *Ripk1* deficiency promoted the basal autophagy in normal condition (**Figure 1j**). We think that under normal conditions, although aspartate levels were increased in *Ripk1*^{-/-} cells, aspartate may have minimal effects on cell survival and autophagy because amino acids are sufficient. However, under starvation conditions, we have demonstrated that the increased aspartate levels inhibited the activation of autophagy and starvation resistance. From this aspect, we agree that *RIPK1* may regulate autophagy by different mechanisms in normal growth conditions and starvation conditions.

The main finding in our study is that *Ripk1* deficiency inhibits the starvation induced autophagy. Therefore, we strongly agreed with reviewer’s comment that “***it would be important to show/measure the level of aspartate upon starvation***”. In the revised manuscript, we further profiled the metabolic changes in starvation conditions in both MEF and Jurkat cells (**Fig. 2d, e**). In addition, metabolic changes in mouse neonate tissues (liver and brain) were also provided (**Fig. 2f, g**). Under starvation conditions, in all of the four data sets, we confirmed that *RIPK1* deficiency caused increased aspartate levels in cells and mouse tissues, which were rescued by *RIPK1* complementation (**Fig. 2h**).

Comment #2: “In Fig. 3 e-l the authors show that reducing the concentration of aspartate via *Got2* siRNA or using the AOA inhibitor in *Ripk1*^{-/-} MEFs allows induction of autophagy upon starvation. This is an important set of data supporting the hypothesis, but it requires some further explanation. Indeed *Got2* siRNA has a significant, but very small effect on the level of aspartate (Fig. 3e) and a very clear effect in inducing autophagy, see for example p62 WB in Fig. 3f; on the contrary, the use of AOA has a massive effect on aspartate (Fig 3i) but a very similar effect on autophagy induction (if not smaller, see for example WB in Fig. 3J). If the level of aspartate inhibits autophagy induction the effect should be proportional to its concentration?”

Ans: We thank the reviewer’s careful examination. The LC-MS data and WB data in original Fig. 3e and 3f, as well as Fig. 3i and 3j were not from the same batch of cell samples. So the change degrees were not well consistent. Especially in Fig. 3e and 3f, the transfection efficiency of MEF cells is low and not consistent in different batches. To solve this issue, in the revised manuscript, we prepared stable GOT2 knockdown cell lines and repeated the LC-MS and WB analyses using the cell samples from the same batch. The new data were provided in **Fig. 3e and 3f** in the revised manuscript (see below). As you see, all of three GOT2 shRNAs successfully reduced the expressions of GOT2 in protein levels. Among them, #1 shRNA induced the most significant decrease of GOT2 level (**Fig. 3f**). Accordingly, we consistently observed the most significant decrease of aspartate level and increase of autophagy in *Ripk1*^{-/-} cells with #1 shRNA (**Fig. 3e**). To conclude, in GOT2 knockdown experiment, the effect that aspartate level inhibits autophagy induction under starvation is proportional to the concentrations of aspartate.

We also repeated the experiments in **Fig. 3i and 3j** using the cell samples from the same batch. As you see, AOA treatment almost completely reduced aspartate levels (<10% left). Although we observed the slightly increased autophagy upon AOA treatment (increased LC3-II levels and decreased p62 levels in lanes 3 and 4), we think the completely reduced aspartate levels may have other side effects to cells. In addition, here, we used relatively high concentrations of AOA (0.5-0.75 mM), which may also have toxicity effects on cells. Therefore, the use of AOA has a massive effect on aspartate, but a relatively similar effect on autophagy induction was observed. We have added the discussion in the revised manuscript.

Copy of Figure 3e, 3f, 3i, 3j

(e) Intracellular aspartate levels in *Ripk1^{-/-}* MEF cells were measured using LC-MS. *Ripk1^{-/-}* MEF cells were transfected with non-target control (NC) or *Got2* shRNAs (#1-3), and cultured in EBSS for 4 h (n = 5 or 6 biologically independent samples per group).

(f) Western blot analyses of LC3, p62 and GOT2 in *Ripk1^{-/-}* MEF cells.

(i) Intracellular aspartate levels in *Ripk1^{-/-}* MEF cells responding to the AOA treatment (0, 0.5 mM, 0.75 mM) in EBSS for 4h were determined by LC-MS (n= 10 biologically independent samples per group).

(j) Western blot analyses of LC3 and p62 levels in *Ripk1^{-/-}* MEF cells responding to the AOA treatment in EBSS for 4 h as indicated.

Comment #3: “The proposed mechanism for the increased level of aspartate in *Ripk1^{-/-}* is inhibition of its catabolism. A number of enzymes involved in aspartate catabolism are shown to be downregulated: GOT1, ASNS, ASS1 and CAD (Fig. 6). These data do support the hypothesis that aspartate catabolism is inhibited, but further evidences are required to fully prove this. For example, what are the levels of the metabolites produced by these reactions? Also, in Fig 5I label aspartate is given to the cells (wt and *Ripk1^{-/-}*), how is it metabolized? Does it accumulate in cells as it should happen if its catabolism is inhibited?”

Ans: We thank the reviewer’s comment. In the revised manuscript, we repeated our [U-¹³C]-aspartate labelling experiment in WT and *Ripk1^{-/-}* MEFs under starvation conditions. Please note that the original labeling experiment was performed under normal condition. As suggested by reviewer, we noticed the importance of starvation condition, and the metabolic changes under starvation in response to *Ripk1* deficiency are more relevant to our study. The new data were provided in **Figure 5i-j** and **Figure 6c** (see below). The results clearly demonstrated that [U-¹³C]-aspartate-derived metabolites in TCA cycles, such as citrate, cis-aconitate, α-ketoglutarate, and malate, were significantly increased in *Ripk1^{-/-}* cells (**Figure 5i-j**). Instead, the catabolism products of [U-¹³C]-aspartate, such as adenylosuccinate and orotate, were significantly decreased in

Ripk1^{-/-} cells (**Fig. 6c**). Together, these results revealed that the catabolism of aspartate in *Ripk1*^{-/-} cells was down-regulated, while the increased levels of aspartate contributed to increase the activity of the TCA cycle.

Finally, as the reviewer pointed out, we also confirmed that [U-¹³C]-aspartate was accumulated in *Ripk1*^{-/-} MEF cells compared to the WT controls.

Copy of Figure 5i, 5j, and 6c

(5i) Schematic illustration of stable isotope tracing using [U-¹³C]-aspartate as a tracer. Cells were cultured with 375 μM [U-¹³C]-aspartate in EBSS for 7 h before collection. Black dots represent ¹²C and red dots represent ¹³C.

(5j) Relative abundances of metabolites in TCA cycle after [U-¹³C]-aspartate treatment of WT and *Ripk1*^{-/-} MEFs for 7 h (n= 9-10 biologically independent samples per group).

(6c) Relative abundances of adenylosuccinate and orotate in aspartate catabolism pathway (n=10 biologically independent samples per group).

Figure R2. The abundances of intracellular [U-¹³C]-aspartate level in WT and *Ripk1*^{-/-} MEF cells.

Comment #4: “The authors should indicate which statistical test have been utilised to calculate significance;”

Ans: We thank the reviewer’s comment. We have added the statistical tests that were used to calculate the significances in revised manuscript.

Comment #5: “There are at least two published manuscripts on the role of *Ripk1* as regulator of metabolism: 10.1126/science.1172308 and 10.1016/j.molcel.2020.11.008. These two papers should be discussed, in particular the second one which is related to the topic covered in this manuscript.”

Ans: We thank the reviewer’s comment. We have added these publications in our discussion.

Comment #6: “Throughout the manuscript experiments are performed alternating normal growth conditions and starvation, not always with a clear rational. It would be great to make the rational clearer and also to indicate clearly in the figure in which conditions the experiment has been performed to help the reader.”

Ans: Thanks for the reviewer's suggestion. We have also noticed the importance of starvation condition for our study, and provided the metabolomics data under starvation conditions and added the details of culture conditions in the revised manuscript.

REVIEWERS' COMMENTS

Reviewer #1 (Remarks to the Author):

The concerns raised regarding the initial submission of this manuscript have all be satisfactorily addressed in the revised manuscript. I now recommend publication of the manuscript.

One suggestion regarding Figure 2. I would suggest eliminating the Venn diagram that leaves aspartate as the only metabolite that is responsive to RIPK1 manipulation. The same metabolites were not assessed in each for each of the 4 data sets so the fact that aspartate is the only one responsive to RIPK1 alterations is not particularly relevant. That being said, the revised data do clearly demonstrate that RIPK1 impacts aspartate (to different degrees) in multiple distinct biological contexts and provide solid justification for the decision to focus subsequent studies on the RIPK1-aspartate connection.

Reviewer #2 (Remarks to the Author):

In the revised version of the manuscript by Mei et al., the authors have addressed my main concerns. In this article, the authors report that RIPK1 regulates starvation resistance through modulating aspartate catabolism. The authors observe that RIPK1 deficiency increases aspartate levels. These increased aspartate levels then enhance TCA cycle and ATP production and ultimately block the AMPK-ULK1 axis of autophagy induction. Furthermore, the authors report that RIPK1 deficiency down-regulates enzymes involved in aspartate catabolism by inactivating the transcription factor SP1. In this revised version, the authors added some additional aspects of this SP1 regulation. I think that this manuscript can be acceptable for publication in NATURE COMMUNICATIONS if some remaining minor aspects can addressed/corrected. In the following, I am referring to my previous numbering in my comments below.

Major point:

- 1) The authors have tried to investigate how RIPK1 regulates the transcription factor SP1. Apparently, nuclear translocation of SP1 is regulated, probably by an altered phosphorylation of SP1. Of note, this is an RIPK1 kinase activity-independent mechanism. I think these observations are important for the paper. I only have two minor points:
 - a. Perhaps the authors can add a quantification for figure 7c
 - b. Perhaps the authors can interpret their observations in the discussion, e.g. in the context of other kinase activity-independent functions of RIPK1.

Minor points:

- 1) ok
- 2) ok
- 3) ok
- 4) ok
- 5) in fact, SLC1A3 not only "not elevated", buter rather decreased in Ripk^{-/-} MEFs. Perhaps the authors can adjust the wording.
- 6) I am still not sure how autophagosomes can be distinguished from autolysosomes in the shown images, but I think this is only a minor aspect.
- 7) ok
- 8) ok
- 9) ok
- 10) ok
- 11) ok
- 12) ok

13) ok

Unfortunately, in the revised version of the paper, some minor errors have been included that were not present in the original version. However, these are only editorial adjustments:

- 1) page 5 (description of figures 1l and 1m): probably it should be "GFP-LC3 puncta were not observed..."
- 2) Supplementary figures S2d and S2e are not mentioned in the text.
- 3) Figure 3m is not mentioned in the text.
- 4) Page 7: I suggest to use "mammalian homolog of yeast Atg1" instead of "mammalian ATG1"
- 5) Page 7: It should probably be "Supplementary Figs. 4d, e" and not "Supplementary Figs. 4e, f"
- 6) Page 10: when referring to Fig. 7e, perhaps the authors can also refer to Supplementary Fig. S7b
- 7) Page 11: It should probably be "Supplementary Fig. 7c, d" and not "Supplementary Fig. 7b, c"
- 8) Page 11: It should probably be "Supplementary Fig. 7e" and not "Supplementary Fig. 7d"
- 9) Page 13: instead of "underlies the" I would suggest "represents the underlying"
- 10) Figure 1g: please include labels LC3 and DAPI
- 11) Figure 5e and 5f: y-axis should be "ECAR"
- 12) Figure 5i and 5j, Figure 6c: replace "labelling" by "labelling"
- 13) Figure S1e: please include labels LC3 and DAPI
- 14) Figure S5g,h: was this also in EBSS (as in figures 5i,j)? Please mention in the legend!

Reviewer #3 (Remarks to the Author):

The revised version of the manuscript by Mei X and colleagues is definitely a stronger version. It is clear that the authors worked hard to address the reviewers' comments and the additional data are a good addition.

With respect of the points raised by this reviewer in the first round. Please see below.

Comment #1. The answer of the author addresses the point, but unfortunately has not been included in the text. Because this is an important point, it should be included in the discussion of the manuscript.

Comment #2. As also recognised by the authors in the rebuttal letter, the data with AOA are not fully convincing. The authors in the rebuttal letter say they have discussed about this in the manuscript, but unfortunately this is nowhere to be found in the manuscript.

Comment #3 #4 #5. The authors have addressed these points.

Comment #6: this point still needs further work. In many part of the text the data are described without specifying in which conditions the experiments were performed, i.e. normal growth conditions vs starvation. This info is provided in the figure legends, but the text is very misleading for the reader who is lead to think many of the data reported where done in normal growth conditions, while the majority had been done after starvation. It will be very important to rephrase the text clearly stating when experiments where performed upon starvation and also to indicate this in the figure themselves (as done for some experiments).

In the text some statements are not supported by data and need rephrasing, see below:

#7: lines 67-68. "However, there were no survival differences in WT or Ripk1-/- cells with and without CHX treatments (Figs. 1a, b)". In the figure the data with CHX only treatment are missing and the condition EBSS + CHX shows differences. Rephrase and/or the data.

#8: lines 87-88. Reference missing?

#9: lines 97-98. "Taken together, these results demonstrated that RIPK1 is essential for cell survival under starvation conditions". No data are shown about cell survival, please rephrase.

#9: lines 113-114. "These results indicated that RIPK1 deficiency caused increased aspartate levels in cells and mouse tissues under starvation, which were rescued by RIPK1 complementation". See also #6 – the differences in aspartate levels were observed also in normal growth conditions, this statement is misleading.

#11: Fig. 6b. Can the authors please comment on the fact that expression of GFP in Ripk1-KO cells rescues Adss and Cad???

#12: lines 230-231. "In WT MEFs, SP1 was largely enriched in the nucleus, while SP1 nuclear translocation was significantly decreased in Ripk1^{-/-} MEFs (Figs. 7b, c). And lines 232-233 "We found that SP1 phosphorylation (T739) was significantly decreased in Ripk1^{-/-} MEFs (Fig. 7d)". Because the cytosolic fraction of SP1 is never visible (Fig. 7b) it is not appropriate to discuss about its translocation. More than anything, SP1 expression seems to increase in the nucleus and the text should be changed accordingly. Fig. 7d shows p-SP1 and while it may appear that SP1 is less phosphorylated, the authors are making a strong statement "that SP1 phosphorylation (T739) was significantly decreased in Ripk1^{-/-} MEFs" and the statement should be tuned down.

Response to the reviewers:

The authors would like to thank the reviewers for the helpful comments. We believed that these comments have strengthened the manuscript considerably.

Reviewer #1:

The concerns raised regarding the initial submission of this manuscript have all be satisfactorily addressed in the revised manuscript. I now recommend publication of the manuscript.

One suggestion regarding Figure 2. I would suggest eliminating the Venn diagram that leaves aspartate as the only metabolite that is responsive to RIPK1 manipulation. The same metabolites were not assessed in each for each of the 4 data sets so the fact that aspartate is the only one responsive to RIPK1 alterations is not particularly relevant. That being said, the revised data do clearly demonstrate that RIPK1 impacts aspartate (to different degrees) in multiple distinct biological contexts and provide solid justification for the decision to focus subsequent studies on the RIPK1-aspartate connection.

Response: We appreciate the positive comments from the reviewer. In the revised manuscript, we have removed the Venn diagram from Figure 2. Accordingly, we also revised the result description in the manuscript as follows:

“By checking all RIPK1-dependent metabolites from mouse brain tissue, liver tissues, MEFs, and Jurkat cells under starvation conditions (Fig. 2d-g), we found that aspartate was the specific metabolite that consistently responded to RIPK1 expression in all four biological sample groups under starvation conditions.”

Reviewer #2:

In the revised version of the manuscript by Mei et al., the authors have addressed my main concerns. In this article, the authors report that RIPK1 regulates starvation resistance through modulating aspartate catabolism. The authors observe that RIPK1 deficiency increases aspartate levels. These increased aspartate levels then enhance TCA cycle and ATP production and ultimately block the AMPK-ULK1 axis of autophagy induction. Furthermore, the authors report that RIPK1 deficiency down-regulates enzymes involved in aspartate catabolism by inactivating the transcription factor SP1. In this revised version, the authors added some additional aspects of this SP1 regulation. I think that this manuscript can be acceptable for publication in NATURE COMMUNICATIONS if some remaining minor aspects can addressed/corrected. In the following, I am referring to my previous numbering in my comments below.

Response: We appreciate the positive comments from the reviewer.

Major point:

1) *The authors have tried to investigate how RIPK1 regulates the transcription factor SP1. Apparently, nuclear translocation of SP1 is regulated, probably by an altered phosphorylation of SP1. Of note, this is an RIPK1 kinase activity-independent mechanism. I think these observations are important for the paper. I only have two minor points:*

a. Perhaps the authors can add a quantification for figure 7c

b. Perhaps the authors can interpret their observations in the discussion, e.g. in the context of other kinase activity-independent functions of RIPK1.

Response: We appreciate the comments from the reviewer. In the revised manuscript, we have added the quantification result for Figure 7c according to the method mentioned in the literature (Mol. Cancer Res., 2014; 12(3): 464-476. doi:10.1158/1541-7786.MCR-13-0398.) Specifically, to quantitatively assess nuclear location of SP1, we first calculated the total areas of green and blue colors in SP1 and DAPI images, respectively. Then, we also calculated the overlapped area between green and blue colors in the merged image. Finally, we calculated the SP1 nuclear location score using the ratio of the overlapping area to the sum of the total green area and the total blue area.

We have added some discussion in the revised manuscript as follows:

“Mechanistically, we found that RIPK1 regulates aspartate metabolism by inhibiting SP1 nuclear expression, probably by an altered phosphorylation (T739) of SP1, in a kinase-independent manner. It has been well reported that some of RIPK1’s functions, such as promotion of cell death, require its kinase activity, whereas others, such as the activation of MAPK and NF-κB to mediate pro-survival signals, are kinase activity independent. Here, our results suggest that kinase activity of RIPK1 is dispensable for starvation resistance and autophagy. Although our data indicated that RIPK1 interacted with SP1 in cells, the mechanism by which RIPK1 regulates SP1 activity and phosphorylation in a kinase-independent manner requires more experimental evidence.”

Minor points:

1) ok

2) ok

3) ok

4) ok

5) *in fact, SLC1A3 not only “not elevated”, but rather decreased in Ripk^{-/-} MEFs. Perhaps the authors can adjust the wording.*

Response: We have revised the "not elevated" to "decreased"

6) *I am still not sure how autophagosomes can be distinguished from autolysosomes in the shown images, but I think this is only a minor aspect.*

Response: We thank the reviewer's comment. In the revised manuscript, we have added the detailed description regarding to the characteristics of autophagosomes and autolysosomes as below:

“Autophagosomes typically have a double membrane. This structure is distinctly visible by TEM as two parallel membrane layers (bilayers) separated by a relatively narrower or wider electron-translucent cleft. Autolysosomes typically have only one limiting membrane. Frequently, they contain electron-dense cytoplasmic material and/or organelles at various stages of degradation. Autolysosomes are typically smaller than autophagosomes, and their structures are typically stained dark in TEM samples.”

7) ok

8) ok

9) ok

10) ok

11) ok

12) ok

13) ok

Unfortunately, in the revised version of the paper, some minor errors have been included that were not present in the original version. However, these are only editorial adjustments:

*1) page 5 (description of figures 1l and 1m): probably it should be “GFP-LC3 puncta were **not** observed...”*

Response: We have changed the sentence as you suggested.

2) Supplementary figures S2d and S2e are not mentioned in the text.

Response: We have mentioned Supplementary Figures 2d and 2e in the main text.

3) Figure 3m is not mentioned in the text.

Response: We have mentioned Figure 3m in the main text.

4) Page 7: I suggest to use “mammalian homolog of yeast Atg1” instead of “mammalian ATG1”

Response: We have changed the “mammalian ATG1” to “mammalian homolog of yeast Atg1” as suggested.

5) Page 7: It should probably be “Supplementary Figs. 4d, e” and not “Supplementary Figs. 4e, f”

Response: We have changed the “Supplementary Figs. 4e, f” to “Supplementary Figs. 4d, e”

6) Page 10: when referring to Fig. 7e, perhaps the authors can also refer to Supplementary Fig. S7b

Response: We confirmed that we have referred Fig.7e and Supplementary Fig. S7b together.

7) Page 11: It should probably be “Supplementary Fig. 7c, d” and not “Supplementary Fig. 7b, c”

Response: We confirmed that “Supplementary Fig. 7c and 7d” is the correct citation.

8) Page 11: It should probably be “Supplementary Fig. 7e” and not “Supplementary Fig. 7d”

Response: We confirmed that we have referred “Supplementary Fig. 7e” in the main text instead of “Supplementary Fig. 7d”.

9) Page 13: instead of “underlies the” I would suggest “represents the underlying”

Response: We have changed the wording to “represents the underlying mechanism”.

10) Figure 1g: please include labels LC3 and DAPI

Response: We have added the labels of LC3 and DAPI in the figure.

11) Figure 5e and 5f: y-axis should be “ECAR”

Response: We have corrected the spelling.

12) Figure 5i and 5J, Figure 6c: replace “labelling” by “labelling”

Response: We have corrected these spellings .

13) Figure S1e: please include labels LC3 and DAPI

Response: We have added the labels of LC3 and DAPI in the figure.

14) Figure S5g,h: was this also in EBSS (as in figures 5i,j)? Please mention in the legend!

Response: The experiments were performed in normal culture condition. We have mentioned this condition in the figure legend as follows:

“MEFs were grown in Dulbecco’s modified eagle medium (DMEM) without glutamine and supplemented 10% dialysed FBS and 4 mM [U-¹³C]-glutamine for 12 h before collection.”

Reviewer #3:

The revised version of the manuscript by Mei X and colleagues is definitely a stronger version. It is clear that the authors worked hard to address the reviewers’ comments and the additional data are a good addition.

Response: We appreciate the positive comments from the reviewer.

With respect of the points raised by this reviewer in the first round. Please see below.

Comment #1. The answer of the author addresses the point, but unfortunately has not been included in the text. Because this is an important point, it should be included in the discussion of the manuscript.

Response: We thank the reviewer’s comment and are sorry for you mistake. In the revised manuscript, we have included the discussion in pages 12-13 which is mentioned as follows:

“It is worth noting that, unlike the inhibition effect of Ripk1 deficiency on starvation-induced autophagy, Ripk1 deficiency promoted the basal autophagy in normal condition (Fig.1j)⁴². We think that, although aspartate levels were increased in Ripk1^{-/-} cells under both normal and starvation conditions, aspartate may have minimal effects on cell survival and autophagy under normal condition since amino acids are sufficient. However, under starvation condition, the increased aspartate levels inhibited the activation of autophagy and starvation resistance. Thus, RIPK1 seems to regulate autophagy by different mechanisms in normal growth condition and starvation condition.”

Comment #2. As also recognised by the authors in the rebuttal letter, the data with AOA are not fully convincing. The authors in the rebuttal letter say they have discussed about this in the manuscript, but unfortunately this is nowhere to be found in the manuscript.

Response: We are sorry for our mistake. In the revised manuscript, we have added the related information in Page 7, which is mentioned as follows:

“Although AOA had a massive effect on aspartate (Fig. 3i), it generated a very similar effect on autophagy induction, which may be due to the side effect of AOA treatment. AOA was reported as an inhibitor of GOT2 as well as an inhibitor of CBS²⁶. In addition, we used relatively high concentrations of AOA for treatment, which may have toxicity effects on cells. ”

Comment #3 #4 #5. The authors have addressed these points.

Comment #6: this point still needs further work. In many part of the text the data are described without specifying in which conditions the experiments were performed, i.e. normal growth conditions vs starvation. This info is provided in the figure legends, but the text is very misleading for the reader who is lead to think many of the data reported where done in normal growth conditions, while the majority had been done after starvation. It will be very important to rephrase the text clearly stating when experiments where performed upon starvation and also to indicate this in the figure themselves (as done for some experiments).

Response: We thank the reviewer’s comment. In the revised manuscript, we have carefully check the main text and figure legends, and make sure that the culture conditions are clearly reported with the data.

In the text some statements are not supported by data and need rephrasing, see below:

#7: lines 67-68. “However, there were no survival differences in WT or Ripk1^{-/-} cells with and without CHX treatments (Figs. 1a, b)”. In the figure the data with CHX only treatment are missing and the condition EBSS + CHX shows differences. Rephrase and/or the data.

Response: We thank the reviewer’s comment. We have rephrased the sentence as “However, under CHX treatment and EBSS starvation condition, Ripk1 deficiency still significantly inhibited cell survival.”

#8: lines 87-88. Reference missing?

Response: Sorry for our mistake, we have added the reference in the revised manuscript.

#9: lines 97-98. "Taken together, these results demonstrated that RIPK1 is essential for cell survival under starvation conditions". No data are shown about cell survival, please rephrase.

Response: We have changed the sentence to "Taken together, these results demonstrated that RIPK1 is essential for starvation resistance under starvation conditions."

#10: lines 113-114. "These results indicated that RIPK1 deficiency caused increased aspartate levels in cells and mouse tissues under starvation, which were rescued by RIPK1 complementation". See also #6 – the differences in aspartate levels were observed also in normal growth conditions, this statement is misleading.

Response: We changed this sentence to "These results indicated that RIPK1 deficiency caused increased aspartate levels in cells and mouse tissues, which were rescued by RIPK1 complementation."

#11: Fig. 6b. Can the authors please comment on the fact that expression of GFP in Ripk1-KO cells rescues Adss and Cad???

Response: We thank the reviewer's comment. We think this might be due to the side effect of plasmids transfection, which accidently promoted the expressions of Adss and Cad in Ripk1^{-/-} MEF cells.

#12: lines 230-231. "In WT MEFs, SP1 was largely enriched in the nucleus, while SP1 nuclear translocation was significantly decreased in Ripk1^{-/-} MEFs (Figs. 7b, c). And lines 232-233 "We found that SP1 phosphorylation (T739) was significantly decreased in Ripk1^{-/-} MEFs (Fig. 7d)". Because the cytosolic fraction of SP1 is never visible (Fig. 7b) it is not appropriate to discuss about its translocation. More than anything, SP1 expression seems to increase in the nucleus and the text should be changed accordingly. Fig. 7d shows p-SP1 and while it may appear that SP1 is less phosphorylated, the authors are making a strong statement "that SP1 phosphorylation (T739) was significantly decreased in Ripk1^{-/-} MEFs" and the statement should be tuned down.

Response: We agreed to the reviewer's comment. In the revised manuscript, we have revised the related statements as follows:

"In WT MEFs, we found that the SP1 expression was largely increased in the nucleus, while nuclear SP1 was significantly decreased in Ripk1^{-/-} MEFs (Figs. 7b, c). SP1 nuclear translocation can be regulated by its phosphorylation³⁶. Accordingly, we also found that phosphorylated SP1 (T739) was significantly decreased in Ripk1^{-/-} MEFs (Fig. 7d)."